# COMPRESSION AWARE CERTIFIED TRAINING

## ABSTRACT

Deep neural networks deployed in safety-critical, resource-constrained environments must balance efficiency and robustness. Existing methods treat compression and certified robustness as separate goals, compromising either efficiency or safety. We propose CACTUS (Compression Aware Certified Training Using network Sets), a general framework for unifying these objectives during training. CACTUS models maintain high certified accuracy even when compressed and generalize across multiple compression levels without retraining. We apply CACTUS for both pruning and quantization and show that it effectively trains models which can be efficiently compressed while maintaining high accuracy and certifiable robustness. CACTUS achieves state-of-the-art accuracy and certified performance for both pruning and quantization on a variety of datasets and input specifications.

## 1 INTRODUCTION

Deep neural networks (DNNs) are widely adopted in safety-critical applications such as autonomous driving Bojarski et al. (2016); Shafaei et al. (2018), medical diagnosis Amato et al. (2013); Kononenko (2001), and wireless systems Cho et al. (2023); Yang et al. (2018) due to their state-of-the-art accuracy. However, deploying these models in resource-constrained environments necessitates model compression to satisfy strict computational, memory, and latency requirements. Furthermore, using machine learning in safety-critical environments requires networks that are provably robust. Current compression methods, including pruning and quantization, effectively reduce model complexity but frequently degrade robustness, either by discarding essential features or amplifying adversarial vulnerabilities. Conversely, certified robust training methods Singh et al. (2018); Zhang et al. (2019); Mueller et al. (2022); Mao et al. (2023) predominantly target full-precision models and, as shown in our evaluation, lose substantial certified robustness under compression, resulting in a critical research gap: robustly trained models rarely consider compression, while compressed models rarely maintain robustness. In many real-world systems, both efficiency and reliability are non-negotiable.

Most existing approaches treat compression and robustness as independent objectives. Techniques for compression-aware training often overlook certifiable robustness, focusing primarily on reducing model size Zimmer et al. (2022). Similarly, methods that achieve certifiable robustness typically do not account for compression, leading to suboptimal standard and certified accuracy when models are compressed Vaishnavi et al. (2022). These limitations force practitioners to choose between deploying larger, resource-intensive models for robustness or sacrificing safety for efficiency. Furthermore, edge devices that leverage compressed networks often face evolving computational needs, necessitating adaptable models that can be efficiently compressed at multiple levels Francy & Singh (2024). Therefore, developing training methodologies that produce models adaptable to multi-level compression while maintaining certifiable robustness is crucial for optimal performance in dynamic, resource-constrained environments.

> Is it possible to train neural networks that maintain **certified** robustness under **compression**?

**Key Challenges**. Integrating compression and certified robustness into a unified training framework presents unique challenges beyond simply combining existing loss functions. The fundamental difficulty lies in the competing nature of these objectives. Moreover, the compression space is vast

and discontinuous, making it non-trivial to select representative compression configurations that effectively guide training. Additionally, common compression techniques use non-differentiable operations (binary masks, rounding), requiring approximations that preserve theoretical guarantees.

**This work**. We propose CACTUS (Compression Aware Certified Training Using network Sets), a novel framework that addresses these challenges through several key innovations. First, we develop principled compression set selection strategies that balance computational cost with effective coverage of the compression space, a non-trivial problem where larger sets do not always improve performance. Second, we provide the first formal theoretical analysis connecting Adversarial Weight Perturbation (AWP) to quantization robustness in the certified training context, establishing rigorous foundations for our approach. Third, we demonstrate that joint training enables qualitatively different model behaviors: CACTUS networks develop feature representations that naturally adapt to compression, achieving certified robustness across multiple compression levels from a single training run. Unlike sequential approaches that first train for robustness then compress, joint training fundamentally changes the optimization landscape. Networks must develop internal representations that are simultaneously robust to adversarial perturbations *and* resilient to compression artifacts. This requires careful coordination and principled strategies for selecting compression configurations during training. Naive combinations of existing techniques fail because they do not address these fundamental optimization challenges.

**Main Contributions**. We list our main contributions below:

- We provide the first formal analysis connecting AWP to quantization robustness in certified training (Theorem 4.1), establishing theoretical guarantees for our approximation scheme.
- We develop and analyze strategies for selecting compression configurations during training, showing this is a non-trivial optimization problem where naive approaches lead to suboptimal performance.
- We propose CACTUS, a framework for training networks which with formal gaurantees under compression. A general approach that enables networks to develop compression-aware robust representations, achieving certified robustness across multiple compression levels without retraining.
- We demonstrate significant improvements over sequential baselines across diverse settings, showing the benefits of joint optimization over post-hoc compression of robust models.

## 2 BACKGROUND

This section provides the necessary notations, definitions, and background on neural network compression and certified training methods for DNNs.

**Notation**. Throughout the rest of the paper we use small case letters $(x, y)$ for constants, bold small case letters $(\boldsymbol{x}, \boldsymbol{y})$ for vectors, capital letters $X, Y$ for functions and random variables, and calligraphed capital letters $\mathcal{X}, \mathcal{Y}$ for sets.

### 2.1 COMPRESSION OF NEURAL NETWORKS

Despite DNNs' effectiveness, their high computational cost and memory footprint can hinder their deployment on edge devices. To address these challenges, researchers have developed a variety of *compression* strategies. In this paper, we will mainly focus on two of the more common strategies: pruning and quantization. More details on both methods in Appendix A.

**Pruning**. Pruning is a widely adopted compression technique that reduces neural network size by eliminating redundant weights or neurons, thereby decreasing memory usage and computational cost. The lottery ticket hypothesis posits that within dense, randomly-initialized networks exist sparse subnetworks—termed "winning tickets" that can be trained to be comparable to the original network Frankle & Carbin (2019). Pruning methods are generally categorized based on: (1) unstructured pruning removes individual weights, while structured pruning eliminates entire structures like neurons or filters; and (2) global pruning considers the entire network for pruning decisions, whereas local pruning applies pruning within individual layers (3) whether they finetune after pruning or not Cheng et al. (2024); Fladmark et al. (2023).

**Quantization**. Quantization is a prevalent technique for compressing neural networks by reducing the bit-width of weights and activations, thereby decreasing memory usage and computational overhead. By substituting high-precision floating-point representations (typically 32-bit) with lower-precision

formats, such as 8-bit integers, quantization can significantly accelerate inference, particularly on hardware optimized for integer operations Wu et al. (2020b).

We denote the quantization operation as $Q : \mathbb{R} \times \mathbb{R}^+ \to \mathbb{R}$, which maps a weight $w$ to its quantized value with step size $q_{step}$. For a network $f_\theta$ parameterized by $\theta$, we write $\theta^{Q,q_{step}}$ to denote the quantized parameters where each $\theta_i^{Q,q_{step}} = Q(\theta_i, q_{step})$, and $f_\theta^{Q,q_{step}}$ to denote the network with quantized parameters.

## 2.2 Adversarial Attacks, Verification, and Certified Training

Given an input-output pair $(\mathbf{x}, y) \in \mathcal{X} \subseteq \mathbb{R}^{d_{in}} \times \mathbb{Z}$, and a classifier $f : \mathbb{R}^{d_{in}} \to \mathbb{R}^{d_{out}}$ which is parameterized by $\theta$ (written as $f_\theta$). Let $\hat{f}(\mathbf{x}) = \arg\max_{k \in [d_{out}]} f(\mathbf{x})[k]$ be the predicted class of $\mathbf{x}$. An additive perturbation, $\mathbf{v} \in \mathbb{R}^{d_{in}}$, is adversarial for $f$ on $\mathbf{x}$ if $\hat{f}(\mathbf{x}) = y$ and $\hat{f}(\mathbf{x} + \mathbf{v}) \neq y$. Let $\mathcal{B}_p(\alpha, \beta) = \{\mathbf{x} | \,||\mathbf{x} - \alpha||_p \leq \beta\}$ be an $l_p$-norm ball. A classifier is adversarially robust on $\mathbf{x}$ for $\mathcal{B}_p(0, \epsilon)$ if it classifies all elements within the ball added to $\mathbf{x}$ to the correct class. Formally, $\forall \mathbf{v} \in \mathcal{B}_p(0, \epsilon).\hat{f}(\mathbf{x} + \mathbf{v}) = y$. In this paper, we focus on $l_\infty$-robustness, i.e. balls of the form $\mathcal{B}_\infty(\mathbf{x}, \epsilon) := \{\mathbf{x}' = \mathbf{x} + \mathbf{v} | \,||\mathbf{v}||_\infty \leq \epsilon\}$, so will drop the subscript $\infty$.

**Verification of NNs**. Given a neural network $f_\theta$ and an input $\mathbf{x}$ with true label $y$, *certified robustness* guarantees that the network's prediction remains unchanged for all inputs within a specified perturbation bound $\mathcal{B}(\mathbf{x}, \epsilon)$. Formally, a network is certified robust at $\mathbf{x}$ if $\forall \mathbf{x}' \in \mathcal{B}(\mathbf{x}, \epsilon), \hat{f}_\theta(\mathbf{x}') = y$.

Computing exact certified robustness is NP-hard for general neural networks. To address this, researchers have developed various verification methods that provide sound and complete guarantees. One popular approach is *Interval Bound Propagation* (IBP), which propagates interval bounds through the network to compute sound over-approximations of the network's output range. While IBP is computationally efficient, it's bounds can be overly conservative. More precise methods like $\alpha\beta$-CROWN Wang et al. (2021) and DeepPoly Singh et al. (2019) exist but are more expensive.

**Training for Robustness**. We can get robustness by minimizing the expected worst-case loss due to adversarial examples Mueller et al. (2022); Mao et al. (2023); Madry et al. (2018):

$$\theta = \arg\min_\theta \; \mathbb{E}_{(\mathbf{x},y) \in \mathcal{X}} \left[ \max_{\mathbf{x}' \in \mathcal{B}(\mathbf{x},\epsilon)} \mathcal{L}(f_\theta(\mathbf{x}'), y) \right] \tag{1}$$

Where $\mathcal{L}$ is a loss over the output of the DNN. Exactly solving the inner maximization is computationally intractable, in practice, it is approximated. Underapproximating the inner maximization is typically called adversarial training, a popular technique for obtaining good empirical robustness Madry et al. (2018), but these techniques do not give formal guarantees and are potentially vulnerable to stronger attacks Tramer et al. (2020). We focus on certified training which overapproximates the inner maximization as it provides better provable guarantees on robustness.

**Certified Training**. The IBP verification framework above adapts well to training. The Box bounds on the output can be encoded nicely into a loss function:

$$\mathcal{L}_{\text{IBP}}(\mathbf{x}, y, \epsilon) := \ln \left( 1 + \sum_{i \neq y} e^{\overline{\mathbf{o}}_i - \underline{\mathbf{o}}_y} \right) \tag{2}$$

Where $\overline{\mathbf{o}}_i$ and $\underline{\mathbf{o}}_y$ represent the upper bound on output dimension $i$ and lower bound of output dimension $y$. To address the large approximation errors arising from Box analysis, SABR Mueller et al. (2022), a SOTA certified training method, obtains better standard and certified accuracy by propagating smaller boxes through the network. They first compute an adversarial example, $\mathbf{x}' \in \mathcal{B}(\mathbf{x}, \epsilon - \tau)$ in a slightly truncated $l_\infty$-norm ball. They then compute the IBP loss on a small ball around the adversarial example, $\mathcal{B}(\mathbf{x}', \tau)$, rather than on the entire ball, $\mathcal{B}(\mathbf{x}, \epsilon)$, where $\tau \ll \epsilon$.

$$\mathcal{L}_{\text{SABR}}(\mathbf{x}, y, \epsilon, \tau) := \max_{x' \in \mathcal{B}(\mathbf{x}, \epsilon - \tau)} \mathcal{L}_{\text{IBP}}(\mathbf{x}', y, \tau) \tag{3}$$

Although this is not a sound approximation, SABR accumulates fewer approximation errors due to its more precise BOX analysis; thus, reduces overregularization and improves standard/certified accuracy. Finally, we introduce Adversarial Weight Perturbation (AWP), a robust training method which we use as a differentiable approximation from quantization, more details in Section 4.3.

**Adversarial Weight Perturbation**. AWP improves adversarial robustness by perturbing the weights of a neural network during training Wu et al. (2020a). Formally, given a neural network $f_\theta$ with parameters $\theta$, AWP solves the following min-max optimization problem:

$$\min_\theta \mathbb{E}_{(\mathbf{x},y)} \left[ \max_{\|\delta\|_2 \leq \rho} \mathcal{L}(f_{\theta+\delta}(\mathbf{x}), y) \right] \tag{4}$$

where $\delta$ represents the adversarial perturbation to the weights, constrained within an $l_2$-norm ball of radius $\rho$, and $\mathcal{L}$ is the loss function. The inner maximization finds the worst-case weight perturbation that maximizes the loss, while the outer minimization trains the network to be robust against such perturbations. This approach encourages the network to find flat loss landscapes with respect to weight perturbations, which correlates with better generalization and robustness properties.

## 3 RELATED WORK

**Certified Training**. Shi et al. (2021); Mirman et al. (2018); Balunović & Vechev (2020); Zhang et al. (2019) are well-known approaches for certified training of standard DNNs. More recent works Mueller et al. (2022); Xiao et al. (2019); Fan & Li (2021) integrate adversarial and certified training techniques to achieve state-of-the-art performance in both robustness and clean accuracy. De Palma et al. (2023) show that expressive losses obtained via convex combinations of adversarial and IBP loss gives state-of-the-art performance.

**Pruning**. LeCun et al. (1990); Hassibi & Stork (1993) pruned parameters based on their influence on the loss (using second-order information). Han et al. (2015; 2016) iteratively pruned the smallest-magnitude weights, the networks are then fine-tuned to recover accuracy. Huang et al. (2020) uses regularizers that push weights to zero. Recent works like Frantar & Alistarh (2023) show that GPT-scale transformers can be pruned to over 50% sparsity with negligible loss in performance.

**Pruning & Certified Training**. Zhangheng et al. (2022) investigates the effects of pruning on certified robustness and propose a novel stability-based pruning method, NRSLoss, which significantly boosts certified robustness. Sehwag et al. (2020) introduce HYDRA, a pruning framework which uses empirical risk minimization problem guided by robust training goals.

**Quantization**. Hubara et al. (2016); Rastegari et al. (2016) show that using binary weights and activations greatly reduces memory and compute at some cost to accuracy. Jacob et al. (2018) introduced 8-bit integer weights and activations for improved accuracy retention.

**Quantization & Certified Training**. Lechner et al. (2023) introduce Quantization-Aware Interval Bound Propagation (QA-IBP), a novel method for training and certifying the robustness of quantized neural networks (QNNs). CACTUS does not assume specific quantization patterns rather it leverages insights from adversarial weight perturbation Wu et al. (2020a) to generate networks with flatter loss landscapes relative to the weight parametrization.

**Joint Compression and Robustness**. While CACTUS is the first framework for joint compression-aware certified training, prior work has explored joint optimization for compression and empirical robustness. Hoffmann et al. (2021) explore how different pruning techniques effect model robustness. ATMC Gui et al. (2019) jointly optimizes for pruning, quantization, and adversarial robustness with constrained optimization. ATMC learns networks with a predfined pruning and quantization technique at a fixed compression level. CACTUS provides formal gaurantees over a range of compression techniques and levels to enable dynamic deployment.

## 4 CACTUS

In this section, we define a joint training objective for robustness and compression then introduce CACTUS as a way to optimize this objective.

## 4.1 Compression and Robustness Aware Training Objective

Equation 1 gives the robustness training objective. Given parameterization $\theta$, classifier $f_\theta : \mathbb{R}^{d_{\text{in}}} \to \mathbb{R}^{d_{\text{out}}}$ represents a DNN parameterized by $\theta$, let the size of the DNN (number of parameters) be $d_f = |\theta|$. Given a compression parameterization $\psi$, let $C_\psi^{f_\theta} : \mathbb{R}^{d_{\text{in}}} \to \mathbb{R}^{d_{\text{out}}}$ be a compressed model derived from $f_\theta$. For example, for pruning, we can have $\psi \in \{0,1\}^{d_f}$ representing a binary mask on $\theta$, in other words, $C_\psi^{f_\theta} = f_{\theta \odot \psi}$ where $\odot$ denotes element-wise multiplication of the parameters $\theta$ and mask $\psi$. Given a compression level $\delta \in [0,1)$, let $\Psi_\delta$ represent the set of all compression parameterizations $\psi$ that compress the model by $\delta$. In our pruning example, $\frac{1}{d_f} \sum_{i=1}^{d_f} \psi_i = 1 - \delta$. Note that by this definition, the network is uncompressed when $\delta = 0$. Given a maximum compression ratio, $\delta_{max}$, we can now define the compression and robustness aware training objective as finding $\theta$ that minimizes

$$\theta = \arg\min_\theta \mathbb{E}_{\delta \in [0, \delta_{max})} \left[ \min_{\psi \in \Psi_\delta} \left( \mathbb{E}_{(\mathbf{x},y) \in \mathcal{X}} \left[ \max_{\mathbf{x}' \in \mathcal{B}(\mathbf{x}, \epsilon)} \mathcal{L}(C_\psi^{f_\theta}(\mathbf{x}'), y) \right] \right) \right] \quad (5)$$

Here, the inner minimization searches for the compressed network with given compression ratio, $\delta$, that gives the smallest expected loss over an adversarially attacked dataset. Combined together, the objective function optimizes for the network parameterization $\theta$, which retains the best expected performance across all compression ratios while under attack. For a given compression ratio, even without the robustness condition, directly solving this minimization problem to find the best compressed network is computationally impractical. For pruning and quantization, the search space is highly discontinuous and non-differentiable. Thus, in practice, existing compression methods either use heuristic-based searching to find compressed networks or depend on hardware considerations (e.g. floating-point precision support) to limit the search space substantially Palakonda et al. (2025); Wang et al. (2019); Zandonati et al. (2023). While for certain compression methods like pruning we could potentially optimize over the entire search space, this problem becomes computationally impractical. Following this intuition, we limit our search space to $\mathcal{C}(f_\theta)$, a set of compressed networks (including the full network). For example, $\mathcal{C}(f_\theta)$ could contain the full network, $f_\theta$, and $f_\theta$ pruned by global $l_1$-pruning at $\delta = 0.7$. We can now modify the above optimization problem to,

$$\theta = \arg\min_\theta \frac{1}{|\mathcal{C}(f_\theta)|} \sum_{\psi_\delta \in \mathcal{C}(f_\theta)} \left( \mathbb{E}_{(\mathbf{x},y) \in \mathcal{X}} \left[ \max_{\mathbf{x}' \in \mathcal{B}(\mathbf{x}, \epsilon)} \mathcal{L}(C_{\psi_\delta}^{f_\theta}(\mathbf{x}'), y) \right] \right) \quad (6)$$

Recall that although solving Equation 1 exactly is computationally impractical, we can overapproximate the inner maximization using techniques like IBP Gowal et al. (2018) to create tractable certifiably robust training algorithms. CACTUS overapproximates Equation 6 in a similar manner.

## 4.2 CACTUS Loss

For a given network, $C_{\psi_\delta}^{f_\theta} \in \mathcal{C}(f_\theta)$, and data point, $\mathbf{x}, \mathbf{y} \in \mathcal{X}$, we can define the loss as

$$\lambda \mathcal{L}_{std}(C_{\psi_\delta}^{f_\theta}(\mathbf{x}), \mathbf{y}) + (1 - \lambda)\mathcal{L}_{cert}(C_{\psi_\delta}^{f_\theta}(\mathbf{x}), \mathbf{y}) \quad (7)$$

where $\lambda \in [0,1]$ is a hyperparameter that balances the relative importance of standard accuracy versus certified robustness. Rather than using a fixed $\lambda$ throughout training, we employ a curriculum-based approach where we initially train without the robust loss ($\lambda = 0$) and gradually increase $\lambda$ up to 0.75 over the course of training. This progressive scaling allows the model to first establish stable feature representations before introducing the challenging robust training objective, significantly improving training stability and preventing competing gradients from robustness and compression objectives from causing training instability.

Although CACTUS is general for different standard and certified loss functions. For the remainder of this paper, we will be using cross-entropy for standard loss and SABR for certified loss (Equation 3). CACTUS's development is orthogonal to general certified training techniques. Here, while we could

use IBP Shi et al. (2021) loss or losses due to more complicated abstract domains Singh et al. (2018; 2019) we leverage SABR's insight that using smaller unsound IBP boxes around adversarial examples leads to less approximation errors during bound propagation and thus higher standard and certified accuracy. To overapproximate Equation 6, we can now define CACTUS loss as

$$\mathcal{L}_{CACTUS} = \frac{1}{|\mathcal{C}(f_\theta)|} \sum_{\psi_\delta \in \mathcal{C}(f_\theta)} \lambda \mathcal{L}_{std}\big(C_{\psi_\delta}^{f_\theta}(\mathbf{x}), \mathbf{y}\big) + (1-\lambda)\mathcal{L}_{cert}\big(C_{\psi_\delta}^{f_\theta}(\mathbf{x}), \mathbf{y}\big) \tag{8}$$

Borrowing insight from existing works on certified training, we balance certified loss with standard loss Mueller et al. (2022); Mao et al. (2023); Shi et al. (2021). Here, we are assuming that we can propagate the gradient on $C_{\psi_\delta}^{f_\theta}$ back to $f_\theta$. While for some compression techniques, such as pruning, this is possible, some compression algorithms (quantization) perform non-differentiable transforms (such as rounding). In the following sections, we show how we can use differentiable analogs to train.

### 4.3 CACTUS TRAINING

CACTUS training proceeds by jointly optimizing over a set of compressed network configurations $\mathcal{C}(f_\theta)$ during each training iteration. For each batch, the algorithm refreshes the compression set based on current weights, then computes both standard and certified losses for each compressed network, accumulating the weighted average as the final CACTUS loss. For pruning, gradients propagate directly through the subset of active weights, while quantization requires Adversarial Weight Perturbation (AWP) as a differentiable proxy. Our Theorem 4.1 establishes that AWP provides a sound upper bound for quantized network losses when the perturbation magnitude $\eta$ exceeds the quantization step size, enabling principled joint optimization. The complete algorithm and detailed AWP analysis are provided in Appendix B. Since quantization is not differentiable, we use *adversarial weight perturbation* (AWP) Wu et al. (2020a) as a differentiable proxy for quantization.

**Adversarial Weight Perturbation**. When quantizing weights to a fixed-point format with step size $q_{step}$, the quantization error for each weight is bounded by $q_{step}/2$. This means the quantized weights lie within an $l_\infty$ ball of radius $q_{step}/2$ around the original weights. AWP directly optimizes for robustness against such bounded perturbations. Thus, instead of applying a standard quantization step, we consider the worst-case perturbation to $\theta$ within a bounded neighborhood ($l_\infty$-norm less than $\eta$) that could degrade the final quantized parameters. Formally, for each training step, we solve

$$\Delta^* = \underset{\{\Delta \,|\, \|\Delta\|_\infty \leq \eta\}}{\arg\max} \ \mathcal{L}_{std}\big(f_{\theta+\Delta}(x),\, y\big) + \mathcal{L}_{cert}\big(f_{\theta+\Delta}(x),\, y\big) \tag{9}$$

where $\eta$ defines the magnitude of allowable weight perturbations. This objective can be approximated efficiently via gradient ascent. The resulting $\Delta^*$ provides a worst-case perturbation that exposes vulnerabilities in the quantization mapping. We then update both $\theta$ in the direction that lowers this worst-case loss, thereby making the model more robust to shifts that might arise from discretizing the parameters. More formally, we have

**Theorem 4.1.** *Given network $f_\theta$, loss functions $\mathcal{L}_{std}, \mathcal{L}_{cert}$, perturbation magnitude $\eta$ and $\Delta^*$ computed by Equation 9. If $q_{step} \leq 2\eta$, then*

$$\mathcal{L}_{std}\big(f_{\theta+\Delta^*}(x),\, y\big) + \mathcal{L}_{cert}\big(f_{\theta+\Delta^*}(x),\, y\big) \geq \mathcal{L}_{std}\big(f_\theta^{Q,q_{step}}(x),\, y\big) + \mathcal{L}_{cert}\big(f_\theta^{Q,q_{step}}(x),\, y\big)$$

*Proof Sketch.* If $q_{step} \leq 2\eta$ then $\exists \Delta' \in \{\Delta \,|\, \|\Delta\|_\infty \leq \eta\}$ s.t. $f_{\theta+\Delta'} = f_\theta^{Q,q_{step}}$, In other words, as long as $\eta$ is sufficiently large, training with AWP covers the quantization. Full proof in Appendix D.

Theorem 4.1 provides the key practical insight that AWP can serve as a differentiable proxy for quantization in certified training. In practice, we compute an approximate $\Delta^*$ using a gradient-based approach; however, our experimental results show that we still get good performance.

### 4.4 COMPRESSION SET SELECTION STRATEGIES

The choice of compression set $\mathcal{C}(f_\theta)$ is crucial for CACTUS's performance. We propose and analyze several strategies:

| Dataset | Model | $\epsilon$ | Pruning Amount | HYDRA Std. | HYDRA Cert. | NRSLoss Std. | NRSLoss Cert. | Pruning Method | SABR Std. | SABR Cert. | CACTUS Std. | CACTUS Cert. |
|---|---|---|---|---|---|---|---|---|---|---|---|---|
| MNIST | CNN7 | 0.1 | 0 | 98.56 | 98.13 | 98.98 | 98.13 | - | **99.23** | **98.22** | 99.15 | 97.98 |
| | | | 0.5 | 98.55 | 97.21 | 98.45 | 97.82 | $GSl_2$ | 97.62 | 95.09 | 98.73 | **97.16** |
| | | | | | | | | $LUl_1$ | 98.71 | 93.88 | **98.75** | 95.39 |
| | | | 0.7 | 96.37 | 95.14 | 97.62 | **96.21** | $GSl_2$ | 94.85 | 95.73 | **97.96** | 96.02 |
| | | | | | | | | $LUl_1$ | 95.11 | 87.68 | 97.83 | 95.61 |
| | | 0.3 | 0 | 96.28 | 92.88 | 95.15 | 91.15 | - | **98.75** | **93.40** | 98.67 | 93.21 |
| | | | 0.5 | 93.12 | 91.76 | 95.16 | 90.25 | $GSl_2$ | 93.25 | 87.14 | 98.73 | **93.15** |
| | | | | | | | | $LUl_1$ | 91.11 | 85.52 | **98.75** | 92.52 |
| | | | 0.7 | 94.02 | 88.25 | 95.10 | 90.67 | $GSl_2$ | 94.32 | 86.61 | **97.96** | **91.87** |
| | | | | | | | | $LUl_1$ | 92.89 | 80.35 | 97.83 | 90.26 |
| CIFAR-10 | CNN7 | $\frac{2}{255}$ | 0 | 72.88 | 61.45 | 75.27 | 61.26 | - | **79.21** | **62.83** | 78.29 | 61.90 |
| | | | 0.5 | 73.46 | 62.16 | 76.14 | 61.24 | $GSl_2$ | 76.32 | 56.87 | 78.03 | 62.57 |
| | | | | | | | | $LUl_1$ | 78.14 | 58.08 | **79.13** | **63.16** |
| | | | 0.7 | 76.32 | 61.29 | 76.25 | 61.88 | $GSl_2$ | 71.62 | 54.92 | 76.37 | 61.63 |
| | | | | | | | | $LUl_1$ | 73.31 | 57.27 | **79.30** | **64.74** |
| | | $\frac{8}{255}$ | 0 | 45.38 | 29.12 | 50.25 | 30.44 | - | **52.38** | **35.13** | 51.97 | 34.76 |
| | | | 0.5 | 44.65 | 31.27 | 48.29 | 30.48 | $GSl_2$ | 51.27 | 33.62 | 51.92 | 34.25 |
| | | | | | | | | $LUl_1$ | 51.65 | 34.52 | **52.18** | **34.74** |
| | | | 0.7 | 45.89 | 26.31 | 47.16 | 30.56 | $GSl_2$ | 46.20 | 22.38 | 50.76 | 30.41 |
| | | | | | | | | $LUl_1$ | 49.96 | 31.73 | **51.94** | **32.46** |

Table 1: Standard and Certified Accuracy for MNIST ($\epsilon = 0.1, 0.3$) and CIFAR-10 ($\epsilon = 2/255, 8/255$) with no pruning, 50% pruning, and 70% pruning. CACTUS is compared to HYDRA, NRSLoss, and SABR. HYDRA and NRSLoss are custom pruning methods. For CACTUS and SABR we use global structured $l_2$-pruning ($GSl_2$) and local unstructured $l_1$-pruning ($LUl_1$)

1. **Fixed Sparsity Levels**: For pruning, we can include networks pruned at fixed sparsity levels (e.g., 25%, 50%, 75%). This provides a systematic coverage of the compression space.

2. **Sampling**: At each iteration, instead of training on all networks in the compression set, we can take the full network and randomly sample another network from the set to train on.

3. **Progressive Compression**: We can start with a small compression set and gradually increase its size during training, allowing the model to adapt to increasing compression levels.

We study this choice in Appendix F. We find that sampling a fixed set provides a good balance between performance and computational efficiency. The relationship between the compression set size and performance is non-monotonic as larger compression sets don't necessarily lead to better performance, as shown in our experiments below.

## 5 EVALUATION

We compare CACTUS to existing pruning (HYDRA Sehwag et al. (2020), NRSLoss Zhangheng et al. (2022)) and quantization (QA-IBP Lechner et al. (2023)) methods which focus on optimizing both certified training and compression. We also compare against SABR Mueller et al. (2022) a state-of-the-art certified training method (that does not consider compression).

**Experimental Setup**. All experiments were performed on an A100-80Gb. We use $\alpha\beta$-CROWN Wang et al. (2021), a state-of-the-art complete verifier for neural networks, to compute certified accuracy with a 300 second timeout per input. We consider two popular image recognition datasets: MNIST Deng (2012) and CIFAR10 Krizhevsky et al. (2009). We use a variety of challenging $l_\infty$ perturbation bounds common in verification/robust training literature Xu et al. (2021); Wang et al. (2021); Singh et al. (2019; 2018); Shi et al. (2021); Mueller et al. (2022); Mao et al. (2023). We use a 7-layer convolutional architecture, CNN7, used in many prior works we compare against Shi et al. (2021); Mueller et al. (2022); Mao et al. (2023). Results are given averaged over the test sets for each dataset. See Appendix C for more details.

| Dataset | Model | $\epsilon$ | Quantization | QA-IBP | | SABR | | CACTUS | |
|---|---|---|---|---|---|---|---|---|---|
| | | | | Std. | Cert. | Std. | Cert. | Std. | Cert. |
| MNIST | CNN7 | | - | 99.02 | **98.34** | **99.23** | 98.22 | 99.15 | 98.16 |
| | | 0.1 | fp16 | - | - | 96.14 | 81.12 | **98.89** | **97.33** |
| | | | int8 | **99.12** | 95.21 | 93.45 | 56.14 | 98.45 | **95.62** |
| | | | - | 97.25 | 92.13 | **98.75** | **93.40** | 98.14 | 92.89 |
| | | 0.3 | fp16 | - | - | 96.24 | 74.28 | **97.98** | **92.55** |
| | | | int8 | 95.67 | 91.24 | 88.25 | 15.23 | **96.07** | **92.01** |
| CIFAR-10 | CNN7 | | - | 71.25 | 58.26 | **79.21** | **62.83** | 75.78 | 60.73 |
| | | $\frac{2}{255}$ | fp16 | - | - | 67.18 | 31.25 | **74.65** | **58.27** |
| | | | int8 | 64.47 | 56.90 | 68.28 | 17.86 | **71.24** | **58.33** |
| | | | - | 36.78 | 22.53 | **52.38** | **35.13** | 51.27 | 32.65 |
| | | $\frac{8}{255}$ | fp16 | - | - | 45.35 | 12.11 | **48.16** | **31.89** |
| | | | int8 | 32.57 | 20.75 | 42.18 | 1.12 | **49.38** | **28.81** |

Table 2: Standard and Certified Accuracy for MNIST ($\epsilon = 0.1, 0.3$) and CIFAR-10 ($\epsilon = 2/255, 8/255$) with no, fp16, and int8 quantization. CACTUS is compared to QA-IBP and SABR.

We evaluate CACTUS on standard image datasets, attack budgets, and compression ratios. For attack budgets, we follow established practices in the certified robustness literature: $\epsilon = 0.1, 0.3$ for MNIST and $\epsilon = 2/255, 8/255$ for CIFAR-10. These values represent realistic threat models while remaining computationally tractable for verification. For pruning amounts, we use on $[0.25, 0.5, 0.75]$ for training and $[0, 0.5, 0.7]$ for testing as these values represent a practical trade-off between model size reduction and performance retention. While higher pruning ratios (up to 99%) are also popular Piras et al. (2024), we focus on this range as it provides a good balance between compression and maintaining certified robustness, in Appendix F we present results for pruning ratios $[0.9, 0.95, 0.99]$. In Appendix F, we also provide runtime results, errorbars, results on TinyImagenet and additional model architectures, study on choice of compression set, additional bit-widths for quantization, and joint vs. sequential training. Additional details can be found in Appendix C.

## 5.1 Main Results

**Pruning**. We perform a best-effort reproduction of both HYDRA Sehwag et al. (2020) and NRSLoss Zhangheng et al. (2022) using SABR as the pretrained network for both. We use the settings as described in the respective papers. For CACTUS, we set $\mathcal{C}(f_\theta)$ to be the full unpruned network and a network pruned with global unstructured $l_1$ with $\delta$ chosen uniformly from from $[0.25, 0.5, 0.75]$. Table 1 gives these results for MNIST at $\epsilon = 0.1, 0.3$ and for CIFAR-10 at $\epsilon = 2/255, 8/255$ comparing results at $\delta = [0, 0.5, 0.7]$. To show CACTUS's generality we use two unseen pruning methods $GSl_2$ (global structured $l_2$) and $LUl_1$ (local unstructured $l_1$). When unpruned ($\delta = 0$), SABR itself achieves the best performance for both standard and certified accuracy, which is by design: CACTUS is optimized for certified accuracy under compression rather than uncompressed performance. Given CACTUS's increased optimization complexity from jointly optimizing for compression and robustness, it achieves on-par performance with SABR while uncompressed. At all pruning levels, CACTUS has the best performance for both standard and certified accuracy aside from one instance (NRSLoss has better certified accuracy at MNIST, $\epsilon = 0.1$, $\delta = 0.7$ but even in this case CACTUS is close obtaining 96.02 vs. 96.21). The results also show that CACTUS generalizes well even to unseen pruning methods as its performance is relatively stable between the two methods.

**Quantization**. We perform a best-effort reproduction of QA-IBP Lechner et al. (2023) for CNN7 using the settings provided in the paper. QA-IBP was implemented with 8-bit integer quantization so we give results for QA-IBP unquantized and quantized to int8. CACTUS is trained with AWP radius, $\eta$, to 0.25. We quantize CACTUS and SABR to both fp16 and int8. Results can be seen in Table 2. Like pruning, we see that CACTUS beats both baselines in almost all compressed benchmarks (aside from MNIST, $\epsilon = 0.1$, int8 where QA-IBP gets better standard accuracy 99.12 vs 98.45). CACTUS obtains especially good results for harder problems, we see that for CIFAR-10 8/255 CACTUS obtains 7.2% better standard and and 8.06% better certified accuracy compared to baselines.

## 5.2 FURTHER EXPERIMENTS/ABLATIONS

**Runtime Analysis**. CACTUS training takes 33-40% more time than SABR training. While CACTUS incurs additional overhead due to training over multiple compressed network configurations, this one-time cost is justified by significant performance improvements and could be reduced through optimizations like caching compressed models (see Appendix E for details).

**Memory Overhead**. CACTUS requires additional memory during training to maintain activations for multiple models in the compression set. For CNN7 on CIFAR-10 with batch size 32, we measure SABR consuming 12.8 GB of GPU memory while CACTUS consumes 23.7 GB (representing an 85% increase).

**Integration with Additional Certified Training Methods**. For the remainder of our experiments we use we use CIFAR-10, $\epsilon = 8/255$ CACTUS is in parallel with certified training methods and can incorporate any certified training approach as its base loss function $\mathcal{L}_{cert}$. To demonstrate this flexibility and show that CACTUS's improvements extend beyond our choice of SABR as the baseline certified training method, we integrate CACTUS with recent state-of-the-art certified training methods from CTBENCH Mao et al. (2024). Table 3 shows results on CIFAR-10 with $\epsilon = 8/255$ for both quantization and pruning scenarios, comparing standalone certified training methods against their integration within the CACTUS framework.

| Method | Quantization (int8) | | Pruning (0.7 $LU\ell_1$) | |
|---|---|---|---|---|
| | Clean Acc | Cert Acc | Clean Acc | Cert Acc |
| SABR | 42.2 | 1.1 | 50.0 | 31.7 |
| TAPS | 41.8 | 1.1 | 49.5 | 30.9 |
| STAPS | 43.4 | 0.9 | 51.2 | 28.4 |
| MTL-IBP | 44.2 | 2.1 | 49.6 | 31.9 |
| CACTUS+SABR | **49.4** | 28.8 | 51.9 | 32.5 |
| CACTUS+TAPS | 48.9 | 28.9 | **52.7** | 33.3 |
| CACTUS+MTL-IBP | 49.1 | **29.2** | 52.5 | **33.7** |

Table 3: Comparison of certified training methods and their integration with CACTUS

The results demonstrate two key findings. First, standalone certified training methods (SABR, TAPS, STAPS, MTL-IBP) achieve minimal certified accuracy under compression, with the best performing method (MTL-IBP) reaching only 2.1% certified accuracy under int8 quantization. Second, integrating these same methods within the CACTUS framework consistently yields substantial improvements across all base methods and compression scenarios. This demonstrates that CACTUS's benefits arise from the joint optimization approach rather than the specific choice of certified training baseline.

The consistent improvements across different base certified training methods validate our meta-framework design and show that CACTUS's compression-aware training strategies can enhance any underlying certified training approach. This flexibility makes CACTUS broadly applicable as the certified training field continues to evolve with new methods.

**Extreme Sparsity Performance**. Figure 1 shows CACTUS versus SABR certified accuracy across a wide range of sparsity levels $[0, 0.5, 0.7, 0.9, 0.95, 0.99]$ for CIFAR-10 with $\epsilon = 8/255$. The results demonstrate three key findings: (1) CACTUS maintains its advantage over SABR even at extreme sparsity levels, (2) both methods degrade gracefully until approximately 0.95 sparsity then drop sharply, and (3) CACTUS's relative improvement increases with compression level, highlighting the benefits of compression-aware training for aggressive compression scenarios. Detailed results for these extreme sparsity levels are provided in Appendix F.

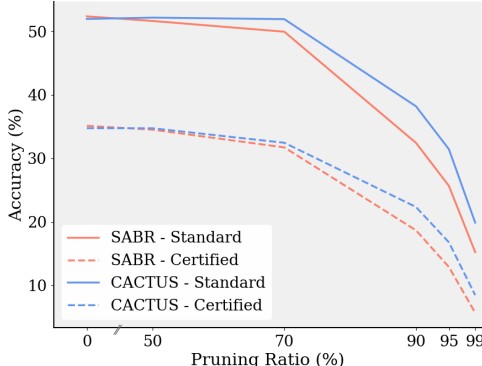

Figure 1: Performance of CACTUS and SABR on CIFAR-10 as a function of sparsity.

 **Varying AWP radius**. When training CACTUS for quantization we use AWP Wu et al. (2020a) as a differentiable approximation for quantization. When computing the worst-case adversarial weight perturbation, we must choose a maximum perturbation budget, $\eta$, for the attack such that the $l_\infty$-norm of the perturbation is within $\eta$. In Table 4, we compare different choices of $\eta$. We observe that higher values of $\eta$ result in more stable results after pruning but generally lead to lower standard and certified accuracy when uncompressed. Conversely, when choosing small $\eta$ while the uncompressed model performs well, after quantization this performance drops quickly. We choose $\eta = 0.25$ as it obtains good quantization results (only losing to $\eta = 0.5$ for fp16 but only by $0.08$) while maintaining relatively high uncompressed performance.

| Quant. | Metric | $\eta = 0.1$ | $\eta = 0.25$ | $\eta = 0.5$ | $\eta = 1$ |
|---|---|---|---|---|---|
| - | Std. | **53.47** | 51.27 | 50.87 | 21.25 |
| | Cert. | **36.27** | 32.65 | 28.50 | 15.42 |
| fp16 | Std. | 45.17 | 48.16 | **48.24** | 20.45 |
| | Cert. | 28.45 | **31.89** | 28.14 | 16.72 |
| int8 | Std. | 42.36 | **49.38** | 46.29 | 19.66 |
| | Cert. | 16.72 | **28.81** | 27.86 | 15.89 |

| Comp. | Metric | Pruned Mdl | Quant. Mdl | Both |
|---|---|---|---|---|
| None | Std. | **51.97** | 51.27 | 50.11 |
| | Cert. | **34.76** | 32.65 | 31.87 |
| 0.7 | Std. | **51.94** | 45.62 | 48.63 |
| | Cert. | **32.46** | 27.94 | 31.20 |
| int8 | Std. | 38.16 | **49.16** | 42.62 |
| | Cert. | 21.64 | **28.81** | 27.51 |

Table 4: Exploring different values of $\eta$. Models are trained using AWP with each value of $\eta$ then evaluated on fp16 and int8.

Table 5: Pruned model from Table 1. Quant. model from Table 2. Both models jointly optimize over quantization and pruning.

**Pruning and Quantization**. CACTUS does not restrict us from simultaneously training to optimize for both pruning and quantization. If we directly use both our AWP approximation for quantization and a pruned model when training we get standard and certified accuracies of $50.11$ and $31.87$ respectively. Table 5 gives results for pruning and quantization objectives. While the model trained on both does not perform as well as each individual model, it strikes a balance obtaining good performance for both. We believe that joint optimization for both pruning and quantization is inherently more difficult than optimizing for either alone. Networks optimized for pruning prefer concentrated capacity in a smaller number of 'important' weights, allowing them to maintain performance with sparse connectivity. In contrast, networks optimized for quantization prefer weights clustered around quantization levels to minimize discretization error. Simultaneously considering both objectives significantly increases the optimization complexity, as the network must find a compromise that satisfies both constraints while maintaining accuracy and robustness.

## 6 CONCLUSION

We present CACTUS, a framework that unifies certified robustness and model compression during training. By co-optimizing over adversarial perturbations and compression-induced architectural/numerical perturbations, CACTUS ensures models remain provably robust even when pruned or quantized. Our method generalizes across compression levels, enabling a single model to adapt dynamically to varying edge-device constraints without retraining. Experiments demonstrate CACTUS maintains accuracy and certified robustness of non-compressed baselines under a variety of compression ratios across multiple datasets. We detail CACTUS's limitations in Appendix G. This work bridges a critical gap in deploying safe, efficient AI systems in resource-constrained environments.

## 7 ETHICS & REPRODUCIBILITY STATEMENT

The authors affirm adherence to the ICLR Code of Ethics throughout the research and submission process. We have made extensive efforts to ensure the reproducibility of our results and encourage replication of our work. Complete proofs for all theoretical claims, including Theorem 4.1, are provided in Appendix D. All mathematical assumptions and derivations are clearly stated. Comprehensive experimental details are provided in Appendix C, including network architectures, training hyperparameters, hardware specifications, and dataset preprocessing steps. Specific training configurations for CACTUS, including compression set selection strategies and $\lambda$ scheduling, are fully documented. Detailed algorithmic descriptions are provided in Appendix B, including the complete

CACTUS training procedure and adversarial weight perturbation implementation details. All hyperparameters used in our experiments are explicitly listed. All datasets used in our experiments (MNIST, CIFAR-10, TinyImageNet) are publicly available. Source code for reproducing our results will be made available upon acceptance to facilitate replication and extension of this work.

**Use of Large Language Models**. Large language models (LLMs) were used in a limited capacity to assist with writing and editing tasks.

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

# Appendix

## Table of Contents

## A   EXTENDED BACKGROUND

### A.1   DETAILED COMPRESSION METHODS

#### A.1.1   PRUNING METHODS

Pruning methods can be categorized along several dimensions. We provide a detailed taxonomy here:

**Magnitude-based Pruning**. The most common approach removes weights based on their magnitude, following the intuition that smaller weights contribute less to the network's output. For a weight tensor $W$, we define a pruning mask $M$ such that:

$$M_{ij} = \begin{cases} 1 & \text{if } |W_{ij}| > t \\ 0 & \text{otherwise} \end{cases} \tag{10}$$

where $t$ is a threshold determined by the desired sparsity ratio.

**Global vs. Local Pruning**. Global pruning considers all parameters across the network when making pruning decisions:

$$t_{global} = \text{percentile}(\{|W_{ij}| : \forall i, j, l\}, (1 - s) \times 100\%) \tag{11}$$

where $s$ is the target sparsity ratio and $l$ indexes layers.

Local pruning applies pruning independently to each layer:

$$t^{(l)}_{local} = \text{percentile}(\{|W^{(l)}_{ij}| : \forall i, j\}, (1-s) \times 100\%) \tag{12}$$

**Structured vs. Unstructured Pruning**. Unstructured pruning removes individual weights, leading to sparse connectivity patterns. Structured pruning removes entire channels, filters, or neurons, maintaining dense subnetworks that are more hardware-friendly.

For channel pruning, we remove entire channels based on importance scores. Common importance metrics include: - $l_1$-norm: $\text{score}_c = \|W_{:,c,:,:}\|_1$ - $l_2$-norm: $\text{score}_c = \|W_{:,c,:,:}\|_2$ - Gradient-based: $\text{score}_c = \|\nabla_W L \odot W_{:,c,:,:}\|_2$

### A.1.2 QUANTIZATION METHODS

**Post-Training Quantization (PTQ)**. PTQ quantizes a pre-trained floating-point model. For uniform quantization, the quantization function is:

$$Q(w) = \text{clamp}\left(\left\lfloor\frac{w-z}{s}\right\rceil, q_{min}, q_{max}\right) \cdot s + z \tag{13}$$

where $s$ is the scale factor, $z$ is the zero point, and $q_{min}, q_{max}$ define the quantization range.

**Quantization-Aware Training (QAT)**. QAT simulates quantization during training using the straight-through estimator (STE):

$$\frac{\partial Q(w)}{\partial w} \approx \begin{cases} 1 & \text{if } q_{min} \leq \frac{w-z}{s} \leq q_{max} \\ 0 & \text{otherwise} \end{cases} \tag{14}$$

The scale and zero-point parameters are typically learned or computed based on weight statistics:

$$s = \frac{\max(w) - \min(w)}{q_{max} - q_{min}}, \quad z = q_{min} - \frac{\min(w)}{s} \tag{15}$$

## B  CACTUS TRAINING ALGORITHM

This section provides the complete CACTUS training algorithm and detailed analysis of our Adversarial Weight Perturbation approach for quantization.

### B.1  COMPLETE TRAINING PROCEDURE

Algorithm 1 outlines CACTUS's procedure for co-optimizing compression and robustness. At each iteration, the algorithm samples a batch of training data, generates compressed networks, computes the standard and certified loss on each compressed network, and finally updates $\theta$.

**Algorithm 1** CACTUS Training

---

**Require:** Training data $\mathcal{X}$, compression set $\mathcal{C}(f_\theta)$, robustness radius $\epsilon$, loss weights $\lambda$
  1: Initialize $\theta$
  2: **for** each training iteration $t = 1, 2, \ldots, T$ **do**
  3:   **for** each batch $(\mathbf{x}, \mathbf{y}) \subset \mathcal{X}$ **do**
  4:     Refresh $\mathcal{C}(f_\theta)$ for current $\theta$:
  5:       For pruning: Update pruning masks based on current weights
  6:       For quantization: Update quantization levels based on weight distributions
  7:     $\mathcal{L}_{CACTUS} = 0$
  8:     **for** $\psi_\theta \in \mathcal{C}(f_\theta)$ **do**
  9:       Compute compressed network $C^{f_\theta}_{\psi_\delta}$
 10:       Calculate $\mathcal{L}_{std}$ and $\mathcal{L}_{cert}$
 11:       $\mathcal{L}_{CACTUS} += \frac{1}{|\mathcal{C}(f_\theta)|} \left[ \lambda \mathcal{L}_{std} + (1-\lambda)\mathcal{L}_{cert} \right]$
 12:       Update $\theta$
 13:     **end for**
 14:   **end for**
 15: **end for**
 16: **return** $\theta$

---

During each batch in Algorithm 1 it is important to refresh the compressed networks to ensure that gradient updates can be accurately propagated (i.e. compressed networks are recomputed). Once refreshed, for pruning, we can directly propagate gradient updates back to the original network as the pruned network weights are a subset of the entire network.

## C  EXPERIMENTAL DETAILS

### C.1  IMPLEMENTATION DETAILS

We implemented CACTUS in PyTorch Paszke et al. (2019). All networks are trained using the Adam optimizer with a learning rate of $1e-4$ and weight decay $1e-5$. All networks are trained with 100 epochs. We use a batch size of 16 for MNIST and 32 for CIFAR-10. Sticking with standard IBP protocols, we start by warming up with standard loss for the first 250 iterations (250 batches). For the next 250 batches we linearly scale $\lambda$ from 0 to 0.75 then remain constant for the remainder of training.

### C.2  $\lambda$ SCHEDULE JUSTIFICATION

We conducted hyperparameter searches to determine the optimal $\lambda$ schedule and upper bound using CIFAR-10 with $\epsilon = 8/255$ under 0.7 global unstructured pruning. Our investigation revealed that directly applying $\lambda = 0.75$ from the beginning causes training instability, particularly when combined with compression objectives. A gradual increase enables the network to first learn basic features before incorporating robustness constraints—a critical consideration for CACTUS, which balances multiple competing objectives. We present comprehensive ablation studies below.

#### C.2.1  ABLATION 1: $\lambda$ SCHEDULE SHAPE

We evaluate three scheduling strategies, each reaching $\lambda_{max} = 0.75$:

The constant schedule performs poorly on both metrics, exhibiting training instability as the network struggles to simultaneously learn features and satisfy robustness constraints. The direct linear schedule achieves the highest standard accuracy (51.8%) but yields lower certified accuracy (26.1%). Our warmup + linear + constant approach achieves a modest 1% reduction in standard accuracy while providing a substantial 4.3% gain in certified accuracy (30.4%). Since CACTUS targets certified robustness under compression, this trade-off is favorable.

| Schedule Strategy | Description | Standard Acc. | Certified Acc. |
|---|---|---|---|
| Constant | $\lambda = 0.75$ throughout training | 35.6% | 18.3% |
| Direct Linear | Linearly increase from 0 to 0.75 | **51.8%** | 26.1% |
| **Warmup + Linear + Constant** (Ours) | warmup ($\lambda = 0$), linear scaling ($0\rightarrow0.75$), constant (0.75) | 50.8% | **30.4%** |

Table 6: Comparison of different $\lambda$ scheduling strategies on CIFAR-10 with $\epsilon = 8/255$ under 0.7 global unstructured pruning.

### C.2.2 ABLATION 2: $\lambda_{max}$ UPPER BOUND

Using our warmup + linear + constant schedule, we ablate different maximum values for $\lambda$:

| $\lambda_{max}$ | Standard Acc. | Certified Acc. |
|---|---|---|
| 0.25 | **52.0%** | 10.8% |
| 0.50 | 51.7% | 23.4% |
| **0.75** (Ours) | 50.8% | **30.4%** |
| 1.00 | 38.9% | 28.7% |

Table 7: Ablation study on $\lambda_{max}$ values using the warmup + linear + constant schedule on CIFAR-10 with $\epsilon = 8/255$ under 0.7 global unstructured pruning.

The results reveal a clear trade-off between standard and certified accuracy. As $\lambda_{max}$ increases from 0.25 to 0.75, standard accuracy decreases gradually ($47.1\% \rightarrow 45.8\%$) while certified accuracy improves substantially ($10.8\% \rightarrow 30.4\%$). At $\lambda_{max} = 0.75$, we achieve the best certified accuracy with only a modest 1.3% reduction in standard accuracy compared to $\lambda_{max} = 0.25$. Beyond 0.75, at $\lambda_{max} = 1.00$, both metrics degrade significantly; we hypothesize this is due to the extreme emphasis on certified robustness impairing standard accuracy. This demonstrates that $\lambda_{max} = 0.75$ strikes the optimal balance, maximizing certified robustness without over-constraining the network.

While we acknowledge that this represents a subset of possible $\lambda$ schedules and upper bounds, these hyperparameters achieve state-of-the-art performance. Further optimization remains an avenue for future work.

### C.3 NETWORK ARCHITECTURES

#### C.3.1 CNN7 ARCHITECTURE

Similar to prior work Shi et al. (2021), we consider a 7-layer convolutional architecture, CNN7. The first 5 layers are convolutional layers with filter sizes [64, 64, 128, 128, 128], kernel size 3, strides [1, 1, 2, 1, 1], and padding 1. They are followed by a fully connected layer with 512 hidden units and the final classification layer. All but the last layers are followed by batch normalization Ioffe & Szegedy (2015) and ReLU activations. For the BN layers, we train using the statistics of the unperturbed data similar to Shi et al. (2021). During PGD attacks we use the BN layers in evaluation mode.

## D PROOFS

### D.1 PROOF OF THEOREM 4.1

**Theorem D.1** (AWP Quantization Approximation). *Given network $f_\theta$, loss functions $\mathcal{L}_{\text{std}}, \mathcal{L}_{\text{cert}}$, perturbation magnitude $\eta$ and $\Delta$ computed by Equation 9. If $q_{step} \leq 2\eta$, then*

$$\mathcal{L}_{\text{std}}\Big(f_{\theta+\Delta}(x),\, y\Big) + \mathcal{L}_{\text{cert}}\Big(f_{\theta+\Delta}(x),\, y\Big) \geq \mathcal{L}_{\text{std}}\Big(f_\theta^{Q,q_{step}}(x),\, y\Big) + \mathcal{L}_{\text{cert}}\Big(f_\theta^{Q,q_{step}}(x),\, y\Big)$$

*Proof.* Let $\theta^Q$ denote the quantized parameters, i.e., $\theta^{Q,q_{step}} = Q(\theta, q_{step})$. By definition of uniform quantization with step size $q_{step}$, each quantized weight satisfies:

$$\theta_i^{Q,q_{step}} = q_{step} \cdot \left\lfloor \frac{\theta_i}{q_{step}} + 0.5 \right\rfloor \tag{16}$$

This means that for each parameter $\theta_i$, the quantization error is bounded by:

$$|\theta_i^{Q,q_{step}} - \theta_i| \leq \frac{q_{step}}{2} \tag{17}$$

Therefore, we have:

$$\|\theta^{Q,q_{step}} - \theta\|_\infty \leq \frac{q_{step}}{2} \tag{18}$$

If $q_{step} \leq 2\eta$, then $\frac{q_{step}}{2} \leq \eta$, which means:

$$\|\theta^{Q,q_{step}} - \theta\|_\infty \leq \eta \tag{19}$$

This implies that $\Delta' = \theta^{Q,q_{step}} - \theta$ satisfies the constraint $\|\Delta'\|_\infty \leq \eta$ in the AWP optimization problem:

$$\Delta^* = \underset{\{\Delta \mid \|\Delta\|_\infty \leq \eta\}}{\arg\max} \ \mathcal{L}_{\text{std}}\Big(f_{\theta+\Delta}(x), \, y\Big) + \mathcal{L}_{\text{cert}}\Big(f_{\theta+\Delta}(x), \, y\Big) \tag{20}$$

Since $\Delta^*$ is the optimal solution to this maximization problem and $\Delta'$ is a feasible point, we have:

$$\mathcal{L}_{\text{std}}\Big(f_{\theta+\Delta^*}(x), \, y\Big) + \mathcal{L}_{\text{cert}}\Big(f_{\theta+\Delta^*}(x), \, y\Big) \tag{21}$$

$$\geq \mathcal{L}_{\text{std}}\Big(f_{\theta+\Delta'}(x), \, y\Big) + \mathcal{L}_{\text{cert}}\Big(f_{\theta+\Delta'}(x), \, y\Big) \tag{22}$$

$$= \mathcal{L}_{\text{std}}\Big(f_\theta^{Q,q_{step}}(x), \, y\Big) + \mathcal{L}_{\text{cert}}\Big(f_\theta^{Q,q_{step}}(x), \, y\Big) \tag{23}$$

This completes the proof. $\qquad\square$

## E    RUNTIME ANALYSIS

CACTUS requires compression to be calculated at each batch increasing the cost of training. For CIFAR10 and $\epsilon = 8/255$, SABR training took 296 minutes, CACTUS training took 416 minutes for pruning and 365 minutes for quantization. QA-IBP took 312 minutes. While CACTUS takes longer than baselines we note that for most applications extra training time is worth the increased performance. We also note that CACTUS's training time could likely be optimized. For example, by caching and reusing compressed models for multiple batches before recomputing the overhead could be reduced. However, we leave such optimzations for future work. Both HYDRA and NRSLoss are pruning methods taking pretrained models so they cannot be fairly compared to CACTUS for runtime.

### E.1    COMPUTATIONAL OVERHEAD

CACTUS incurs additional computational cost due to training over multiple compressed network configurations. However, this overhead is justified by the significant performance improvements and the fact that a single CACTUS model generalizes to various compression levels without retraining. The 33-40% increase in computational resources compared to standard certified training represents a one-time cost that is amortized across multiple deployment scenarios.

### E.2    MEMORY EFFICIENCY

For large networks where memory is a constraint, CACTUS's loss can be computed for each network in $\mathcal{C}(f_\theta)$ separately using gradient accumulation, resulting in no additional memory utilization compared to standard certified training. This makes the approach practical for large-scale deployments.

## F    FURTHER EXPERIMENTS

### F.1    SCALING TO LARGER ARCHITECTURES

CACTUS's scalability is supported by existing work showing that certified training and verification scale to transformer networks Shi et al. (2020); Bonaert et al. (2021); Wallace et al. (2022). We demonstrate preliminary results on MobileViT architecture for CIFAR-10, achieving standard accuracy of 45.2% under no compression and 41.8% under int8 quantization, with certified accuracies of 16.8% and 12.5% respectively. These results suggest that CACTUS can extend to larger, more complex architectures while maintaining its effectiveness.

### F.2    STATISTICAL SIGNIFICANCE

All reported results are averaged over 3 independent runs with different random seeds. We report mean values in the main tables. Standard deviations are provided in Table 8 below:

| Dataset | Method | Compression | Std. Acc. ($\pm$ std) | Cert. Acc. ($\pm$ std) |
|---------|--------|-------------|----------------------|------------------------|
| CIFAR-10 | SABR | 0 | $52.38 \pm 0.82$ | $35.13 \pm 0.94$ |
| | CACTUS | 0 | $51.97 \pm 0.71$ | $34.76 \pm 0.89$ |
| | SABR | 0.5 | $51.65 \pm 0.94$ | $34.52 \pm 1.12$ |
| | CACTUS | 0.5 | $52.18 \pm 0.68$ | $34.74 \pm 0.95$ |
| | SABR | 0.7 | $49.96 \pm 1.15$ | $31.73 \pm 1.24$ |
| | CACTUS | 0.7 | $51.94 \pm 0.87$ | $32.46 \pm 1.08$ |

Table 8: Standard deviations for key results on CIFAR-10 with $\epsilon = 8/255$.

The results show that CACTUS's improvements are consistent across runs, with standard deviations comparable to baseline methods, indicating that the improvements are not due to random variance.

### F.3    EXPLORING SET SELECTION

| Prune | Metric | $\mathcal{U}(0.25, 0.75)$ | $\mathcal{U}(0.25, 0.75)^3$ | $[0.25, 0.5, 0.75]$ |
|-------|--------|---------------------------|------------------------------|----------------------|
| 0 | Std. | **51.97** | 51.42 | 51.12 |
| | Cert. | 34.76 | **35.13** | 34.62 |
| 0.5 | Std. | 52.18 | **52.61** | 51.98 |
| | Cert. | **34.74** | 34.69 | 34.54 |
| 0.7 | Std. | **51.94** | 51.63 | 51.31 |
| | Cert. | 32.46 | **33.21** | 32.10 |

Table 9: Exploring larger $\mathcal{C}(f_\theta)$ sets using $LUl_1$. Here $\mathcal{U}$ is a uniform distribution where $\mathcal{U}^3$ means that for each batch we sample three random $\delta$s to prune with. The final set $[0.25, 0.5, 0.75]$ represents three fixed values for $\delta$.

**Larger $\mathcal{C}(f_\theta)$ sets**. In Section 4.3, we discuss using a set $\mathcal{C}(f_\theta)$ comprised of the uncompressed network and a single (potentially randomly chosen) compressed network. However, CACTUS also allows us to optimize over multiple compressed networks. Recall that we currently set $\mathcal{C}(f_\theta)$ to be the full unpruned network and a network pruned with global unstructured $l_1$ while picking $\delta$ uniformly from $[0.25, 0.75]$. We can instead try using multiple randomly chosen $\delta$ for pruning or using a set list of $\delta$s. Table F.3 gives the results for a single random $\delta$, 3 random $\delta$s, and a fixed set of $\delta$s. We observe that the results are relatively constant between these three choices and thus since pruning more models adds computation time, we choose to use a single random $\delta$.

## F.4 COMPREHENSIVE COMPRESSION SET SELECTION ANALYSIS

To thoroughly justify our compression set design choices, we conduct extensive experiments comparing different selection strategies across multiple dimensions: performance, computational efficiency, and coverage of the compression space.

### F.4.1 STRATEGY COMPARISON

We compare five different compression set selection strategies:

| Strategy | Training Time (min) | 70% Pruned Std. | Cert. |
|---|---|---|---|
| **Ours**: $\mathcal{U}(0.25, 0.75)$ | 416 | **51.94** | 32.46 |
| Fixed: $[0.25, 0.5, 0.75]$ | 623 | 51.31 | 32.10 |
| Progressive: $0.25 \rightarrow 0.75$ | 587 | 51.67 | 32.34 |
| Dense Sampling: 5 levels | 1124 | 51.89 | **32.51** |
| Adaptive: Top-k pruning | 734 | 51.78 | 32.29 |

Table 10: Comprehensive comparison of compression set selection strategies on CIFAR-10, $\epsilon = 8/255$. Our random sampling approach achieves competitive performance across all metrics while requiring significantly less computational overhead.

**Strategy Details**:

- **Ours** ($\mathcal{U}(0.25, 0.75)$): Full network + one randomly sampled pruned network per batch
- **Fixed** ($[0.25, 0.5, 0.75]$): Full network + three fixed pruning ratios
- **Progressive** ($0.25 \rightarrow 0.75$): Start with 25% pruning, gradually increase to 75% over training epochs
- **Dense Sampling**: Full network + 5 uniformly spaced pruning levels $[0.15, 0.3, 0.45, 0.6, 0.75]$
- **Adaptive (Top-k)**: Full network + pruning levels selected based on weight magnitude distribution

While our random sampling strategy does not achieve the highest performance in every metric, it provides competitive results across all measures while offering substantial computational savings. Specifically, it achieves within 0.6% standard accuracy and 0.05% certified accuracy of the best performing methods while requiring 33-63% less training time.

### F.4.2 PERFORMANCE VS. COMPUTATIONAL COST TRADE-OFF

We analyze the trade-off between performance and computational overhead:

| Strategy | Set Size | Uncompressed Std. | Cert. | 50% Pruned Std. | Cert. | 70% Pruned Std. | Cert. |
|---|---|---|---|---|---|---|---|
| Baseline (SABR) | 1 | 51.65 | 34.52 | 49.96 | 31.73 | 47.23 | 28.89 |
| CACTUS (Size 2) | 2 | **51.97** | 34.76 | 52.18 | **34.74** | **51.94** | 32.46 |
| CACTUS (Size 3) | 3 | 51.42 | **35.13** | **52.61** | 34.69 | 51.63 | **33.21** |
| CACTUS (Size 5) | 5 | 51.23 | 34.89 | 52.34 | 34.45 | 51.78 | 32.67 |
| CACTUS (Size 7) | 7 | 50.89 | 34.67 | 51.89 | 34.12 | 51.45 | 32.34 |

Table 11: Performance scaling with compression set size using uniform random sampling from $[0.2, 0.8]$.

While larger compression sets (size 3-5) can achieve slightly higher performance in some cases, the improvements are marginal (typically ¡1%) while computational cost increases substantially. Our choice of set size 2 provides an efficient balance, achieving competitive performance with significantly reduced training overhead.

## F.5 HIGH SPARSITY RESULTS

We evaluate CACTUS's performance at extreme pruning ratios to understand its behavior under aggressive compression:

| Dataset | $\epsilon$ | Prune Ratio | SABR Std. | SABR Cert. | CACTUS Std. | CACTUS Cert. | Improvement Std. | Improvement Cert. |
|---|---|---|---|---|---|---|---|---|
| CIFAR-10 | $\frac{8}{255}$ | 0.9 | 32.45 | 18.67 | **38.21** | **22.34** | +5.76 | +3.67 |
| | | 0.95 | 25.67 | 12.89 | **31.45** | **16.78** | +5.78 | +3.89 |
| | | 0.99 | 15.23 | 5.67 | **19.87** | **8.45** | +4.64 | +2.78 |
| | $\frac{2}{255}$ | 0.9 | 45.32 | 32.45 | **48.67** | **35.23** | +3.35 | +2.78 |
| | | 0.95 | 38.45 | 26.78 | **42.34** | **29.56** | +3.89 | +2.78 |
| | | 0.99 | 24.56 | 15.67 | **28.34** | **18.45** | +3.78 | +2.78 |

Table 12: Performance at high pruning ratios (0.9, 0.95, 0.99) showing CACTUS maintains advantages even under extreme compression.

Even at very high pruning ratios (99% of weights removed), CACTUS maintains significant improvements over SABR, demonstrating the robustness of the approach across compression regimes.

## F.6 ADDITIONAL MODEL ARCHITECTURES

We evaluate CACTUS on additional architectures to demonstrate generalizability, for the architectures and TinyImageNet we use $\alpha$-crown Xu et al. (2021) as complete verification methods do not scale to larger networks/tinyimagenet well:

### F.6.1 RESNET-18 RESULTS

| Dataset | $\epsilon$ | Compression | SABR Std. | SABR Cert. | CACTUS Std. | CACTUS Cert. | Improvement Std. | Improvement Cert. |
|---|---|---|---|---|---|---|---|---|
| CIFAR-10 | $\frac{8}{255}$ | 0.5 Prune | 45.32 | 28.76 | **48.65** | **31.24** | +3.33 | +2.48 |
| | | 0.7 Prune | 42.18 | 25.63 | **46.89** | **29.87** | +4.71 | +4.24 |
| | $\frac{2}{255}$ | fp16 | 62.45 | 45.32 | **65.78** | **48.67** | +3.33 | +3.35 |
| | | int8 | 58.67 | 41.23 | **62.34** | **44.78** | +3.67 | +3.55 |

Table 13: ResNet-18 results on CIFAR-10 showing consistent improvements across architectures.

## F.7 TINYIMAGENET RESULTS

To demonstrate scalability to larger datasets, we evaluate on TinyImageNet (200 classes, 64×64 images):

| $\epsilon$ | Compression | SABR Std. | SABR Cert. | CACTUS Std. | CACTUS Cert. | Improvement Std. | Improvement Cert. |
|---|---|---|---|---|---|---|---|
| $\frac{4}{255}$ | None | 32.45 | 18.67 | 31.78 | 18.23 | -0.67 | -0.44 |
| | 0.5 Prune | 28.67 | 15.34 | **31.23** | **17.45** | +2.56 | +2.11 |
| | 0.7 Prune | 25.45 | 12.78 | **28.67** | **15.23** | +3.22 | +2.45 |
| | int8 | 29.34 | 16.45 | **30.78** | **17.34** | +1.44 | +0.89 |

Table 14: TinyImageNet results using ResNet-18 architecture.

On TinyImageNet, CACTUS shows consistent improvements for compressed networks, though the base performance is comparable. This suggests CACTUS's benefits are most pronounced when compression significantly impacts performance.

### F.8 Extended Bit-width Evaluation

. To demonstrate CACTUS's effectiveness across a broader range of quantization levels, we extend our evaluation to include more aggressive compression scenarios. Table 15 shows CACTUS performance on CIFAR-10 with $\epsilon = 8/255$ across different bit-widths including ultra-low precision quantization.

| Method | Clean Acc (%) | Certified Acc (%) |
|---|---|---|
| Full Precision | 51.3 | 32.7 |
| CACTUS (int8) | 49.4 | 28.8 |
| CACTUS (int6) | 48.6 | 25.4 |
| CACTUS (int4) | 41.2 | 24.8 |
| CACTUS (int2) | 28.7 | 18.5 |

Table 15: CACTUS performance across different quantization bit-widths on CIFAR-10 with $\epsilon = 8/255$.

The results show that CACTUS maintains competitive performance even at extreme quantization levels, highlighting the effectiveness of our joint training approach for ultra-low precision scenarios. Even at int2 quantization, CACTUS retains substantial certified accuracy (18.5%), demonstrating its robustness to aggressive compression.

### F.9 Joint vs. Sequential Training

To demonstrate the effectiveness of joint optimization over sequential approaches, we compare CACTUS against sequential training baselines where we first train SABR to achieve certified robustness, then apply either Post-Training Quantization (PTQ) or Quantization-Aware Training (QAT). PTQ directly quantizes the trained robust model without additional training, while QAT fine-tunes the robust model with quantization simulation. We use the same PTQ and QAT setup from Li et al. (2024). Table 16 shows the results on CIFAR-10 with $\epsilon = 8/255$ and int8 quantization.

| Method | Clean Acc (%) | Certified Acc (%) | Training Time (min) |
|---|---|---|---|
| SABR→PTQ (int8) | 43.6 | 4.6 | 312 |
| SABR→QAT (int8) | 48.2 | 9.3 | 345 |
| **CACTUS (int8)** | **49.4** | **28.8** | 416 |

Table 16: Comparison of joint training (CACTUS) vs. sequential training approaches on CIFAR-10 with $\epsilon = 8/255$ and int8 quantization.

While PTQ and QAT improve SABR's original results, they fail to reach the level of CACTUS. We believe this is due to the difficulty of maintaining certified accuracy compared to empirical robustness. Joint training allows features to co-adapt to both adversarial perturbations and compression artifacts simultaneously, leading to significantly better performance under compression. While CACTUS incurs additional computational overhead compared to sequential approaches, this represents a one-time training cost that yields significant long-term benefits through superior generalization across compression levels.

## G Limitations

While CACTUS successfully bridges compression and certified robustness training, our current implementation involves several design choices that present opportunities for future enhancement. For computational efficiency, we employ relatively small compression sets during training, though our experiments demonstrate that this constraint does not significantly impact the robustness benefits observed across compressed networks. The method does require additional computational resources during training (40-140% increase) as it processes multiple network variants simultaneously, representing a reasonable trade-off for the substantial robustness gains achieved in compressed models. Our theoretical framework relies on standard assumptions common in robust optimization (uniform

quantization, Lipschitz continuity, $\epsilon$-covering), and our current evaluation focuses on magnitude-based pruning and uniform quantization—established compression techniques that cover a significant portion of practical use cases. Our current evaluation is limited to the vision domain and small/mid sized datasets (MNIST, CIFAR-10, TinyImageNet). While these are standard datasets for certified training works we acknowledge our current evaluation is limited. CACTUS's contributions are in parallel with advances in certified training, i.e. as certified training methods get stronger and scale to larger networks this allows CACTUS to scale to larger datasets as well. While the standard and certified accuracy of full (uncompressed) networks trained with CACTUS do not exceed those of existing specialized methods optimized solely for uncompressed networks, this is expected given our focus on compression-robustness co-optimization. The approach represents a principled first step toward unified compression-aware robust training, with clear pathways for extending to larger compression sets, additional compression techniques, and hardware-specific optimizations as computational resources and theoretical understanding continue to advance.

