# OpenReview forum: "Compression Aware Certified Training"
_ICLR.cc/2026/Conference — Submitted to ICLR 2026_

### Official Review · Reviewer_AbHd · 2025-10-31

**Soundness:** 1
**Presentation:** 2
**Contribution:** 1
**Rating:** 2
**Confidence:** 4

**Summary:**

This paper introduces CACTUS, a framework for unifying optimization techniques to achieve certified robustness in compressed models. It proposes a CACTUS loss to balance standard accuracy and robustness on a compressed model. While it is the first work to address certified robustness for model compression, its optimization approach is not well-motivated, lacks deep theoretical analysis, and the presentation is not good.

**Strengths:**

1. This paper is the first work to address certified robustness for model compression.
2. The background section is well-written, providing a clear foundation to understand the problem.

**Weaknesses:**

1. **Writing and Notation Quality**. The paper suffers from many writing and notation issues that significantly affect readability and clarity.
- Several key symbols are used before being defined, such as $\theta$, $Q(\cdot)$, $\Delta$, and $\eta$.
- The notation convention stated in Section 2 is inconsistent with later usage. For example, the authors claim lowercase bold letters denote vectors, but $\theta$, which may represent network parameters, should arguably be bold ($\boldsymbol{\theta}$) under that rule.
- The definition of $f$ is **ambiguous and inconsistent**. In Section 2.2, $f_k(\mathbf{x})$ denotes the score for class $k$, but later $f_\theta$ refers to a parameterized model. The paper never explicitly defines $f_\theta(\mathbf{x})$, though it is used in Equation 1. It is unclear whether $f$ denotes a function, a variable, or model logits. This confusion propagates through the theoretical and algorithmic sections, making the paper unnecessarily difficult to follow.
- missing “{”, “}” in the experiment section.

2. **Theoretical Analysis (Theorem 4.1) Is Trivial and Problematic**
- Theorem 4.1 should used  $\Delta^*$ instead of   $\Delta$.
- Theorem 4.1 is almost tautological. Since $\Delta^*$ in Eq. 9 is defined as the optimal perturbation maximizing the loss within the admissible space, it is naturally larger than any feasible perturbation such as that induced by quantization. Hence, the inequality does not provide real theoretical insight.
- The proof in the appendix uses the condition $q_{\text{step}} < 2\eta$, while the main text lists $q_{\text{step}} < \eta$. This inconsistency undermines the claimed theoretical guarantee.

3. **Computational Overhead of CACTUS Loss**
The CACTUS objective (Eq. 8) involves averaging losses over a set of compressed models, $C(f_{\theta})$. In practice, this set can be extremely large because multiple compression configurations can satisfy the same compression ratio. The approach implies training many compressed models simultaneously, resulting in high computational and memory overhead. The paper does not quantify this cost or justify its scalability.

**Questions:**

1. The paper briefly mentions in the experimental setup that “we set $C(f_{\theta})$ to be the full unpruned network and a network pruned ..." Does this mean that $C(f_{\theta})$ only includes two models (the full and one pruned network)? Theoretically, $C(f_{\theta})$ should contain a large number of compressed models even for one pruning ratio. Could the authors clarify how $C(f_{\theta})$ is actually collected or sampled in practice, and whether the set is fixed or dynamically updated during training?
2. How is certified accuracy computed in experiments?
3. The paper states that $\lambda$ is gradually increased from 0 to 0.75 during training. What motivates the choice of the upper bound 0.75? Did the authors conduct ablation studies on different $\lambda$ schedules or maximum values?

---

> ### Author Response · Authors · 2025-11-21
>
> We thank Reviewer AbHd for acknowledging that CACTUS is "the first work to address certified robustness for model compression" and that our "background section is well-written." We take the concerns about presentation and theoretical analysis very seriously and will make substantial revisions to address all points raised.
>
> > Writing and Notation Quality
>
> We apologize for the notation issues and have thoroughly revised the paper to fix all these problems. We have added definitions to variables that were missing before.
>
> We now consistently use bold font for vectors. We have also fixed our ambiguous $f$ definition by removing $f_k(\mathbf{x})$ and clearly defining $f_\theta$ in Section 2 (Background). We have carefully proofread and fixed all LaTeX syntax errors including the missing braces in Section 5 (Evaluation).
>
> > Theorem 4.1: Theoretical Concerns
>
> We appreciate the detailed technical critique and have addressed each point. First, you are correct that the theorem should use $\Delta^*$ instead of $\Delta$, and we have fixed that in Theorem 4.1 (Section 4).
>
> Second, regarding your concern that the theorem is "almost tautological" and "does not provide real insight," we respectfully disagree with this characterization but understand why it might appear this way. The insight that "optimizing for larger perturbation set bounds smaller perturbation outcomes" is a standard technique, but applying it to connect AWP and quantization in certified training is novel to our knowledge. The main contribution is the insight that AWP can serve as a differentiable proxy for quantization in certified training, with the theorem and proof serving as theoretical motivation for this choice. We have reframed Theorem 4.1's presentation in Section 4 (Technical Contribution) to emphasize the practical insight.
>
> Third, regarding the proof condition inconsistency between $q_{step} < 2\eta$ and $q_{step} < \eta$, thank you for pointing this out. The correct condition is $q_{step} \leq 2\eta$ (equivalently, $\eta \geq q_{step}/2$), and this has been corrected in Theorem 4.1 (Section 4).
>
> > Computational Overhead of CACTUS Loss and $\mathcal{C}(f_{\boldsymbol{\theta}})$ choice in practice
>
> We have added a runtime analysis discussion in Section 5.2, which refers to our longer discussion in Appendix E. In summary, we find that CACTUS takes 33-40% more time to train than SABR. For the final version of the paper, we will also include a section on memory overhead, but due to time limitations in the rebuttal period, we were not able to rerun all of our experiments while measuring memory overhead. We reran the CIFAR-10 pruning experiment with $\epsilon$ = 8/255 and found that SABR consumed a maximum of 12.8 GB of GPU memory while CACTUS consumed 23.7 GB, representing an 85% increase in GPU memory (still a small fraction of what training modern LLMs takes).
>
> While we agree that $C(f_\theta)$ can be large in practice, our experimental results show that our method of taking the full network and a randomly sampled network from our set provides a good balance of performance and cost. In Section 4.4, we discuss different compression set selection methods tried, and Appendix F.3 and F.4 present our ablation results, which indicate that our sampling strategy of sampling a single compression parameterization per batch from a fixed set is effective.
>
> We also note that CACTUS's contributions are in parallel with advances in certified training, i.e. as certified training methods get stronger and scale to larger networks this allows CACTUS to scale to larger datasets/networks as well. In particular, recent works have shown that verification can scale to transformer networks [1,2] and we expect that certified training approaches will soon follow. Finally, we believe the additional runtime overhead is acceptable given CACTUS's performance gains over baselines. In our discussion with Reviewer 7Fvw, we further highlight that CACTUS's focus is on certified network robustness under compression. CACTUS is a one-time training cost that can be used dynamically for different compression techniques and levels, making it more efficient than training separate compressed networks from scratch each time. Furthermore, if memory is a bottleneck for CACTUS, we can use tricks like gradient accumulation or checkpointing to effectively reduce memory overhead at the cost of some additional runtime.
>
> > How is certified accuracy computed in experiments?
>
> We use $\alpha, \beta$-CROWN [3] (with a 300 second timeout), which is the state-of-the-art complete verifier for neural networks. Certified accuracy is computed as (number certified robust and correctly classified examples) divided by (total test examples). We have updated the experimental setup section to address this.

---

> ### Author Response · Authors · 2025-11-21
>
> > Why $\lambda$ = 0.75 upper bound? Ablation studies on $\lambda$ schedule?
>
> We conducted hyperparameter searches to determine the optimal $\lambda$ schedule and upper bound using CIFAR-10 with $\epsilon=8/255$ under 0.7 global unstructured pruning. Our investigation revealed that directly applying $\lambda = 0.75$ from the beginning causes training instability, particularly when combined with compression objectives. A gradual increase enables the network to first learn basic features before incorporating robustness constraints—a critical consideration for CACTUS, which balances multiple competing objectives. We have expanded this analysis with comprehensive ablation studies presented below.
>
> **Ablation 1:** $\lambda$ Schedule: We evaluate three scheduling strategies, each reaching $\lambda_{max} = 0.75$:
>
> | Schedule Strategy | Description | Standard Acc. | Certified Acc. |
> |-------------------|-------------|---------------|----------------|
> | Constant | $\lambda = 0.75$ throughout training | 35.6% | 18.5% |
> | Direct Linear | Linearly increase from 0 to 0.75 | **51.8%** | 26.1% |
> | **Warmup + Linear + Constant** (Ours) | warmup ($\lambda=0$), linear scaling (0→0.75), constant (0.75) | 50.8% | **30.4%** |
>
> The constant schedule performs poorly on both metrics, exhibiting training instability as the network struggles to simultaneously learn features and satisfy robustness constraints. The direct linear schedule achieves the highest standard accuracy (51.82%) but yields lower certified accuracy (26.11%). Our warmup + linear + constant approach achieves a modest 1% reduction in standard accuracy while providing a substantial 4.3% gain in certified accuracy (30.41%). Since CACTUS targets certified robustness under compression, this trade-off is favorable.
>
> **Ablation 2: $\lambda_{max}$ Upper Bound**: Using our warmup + linear + constant schedule, we ablate different maximum values for $\lambda$:
>
> | $\lambda_{max}$ | Standard Acc. | Certified Acc. |
> |-----------------|---------------|----------------|
> | 0.25 | **52.1%** | 10.8% |
> | 0.50 | 51.7% | 23.4% |
> | **0.75** (Ours) | 50.8% | **30.4%** |
> | 1.00 | 38.9% | 28.7% |
>
> The results reveal a clear trade-off between standard and certified accuracy. As $\lambda_{max}$ increases from 0.25 to 0.75, standard accuracy decreases gradually (47.1% → 45.8%) while certified accuracy improves substantially (10.8% → 30.4%). At $\lambda_{max} = 0.75$, we achieve the best certified accuracy with only a modest 1.3% reduction in standard accuracy compared to $\lambda_{max} = 0.25$. Beyond 0.75, at $\lambda_{max} = 1.00$, both metrics degrade significantly; we hypothesize this is due to the extreme emphasis on certified robustness impairing standard accuracy. Therefore, we use $\lambda_{max} = 0.75$ as it has the best balance of standard and certified accuracy among the $lambda$'s we try, maximizing certified robustness without over-constraining the network.
>
> We have incorporated these ablations into Appendix C.2. While we acknowledge that this represents a subset of possible $\lambda$ schedules and upper bounds, we note that these hyperparameters achieve state-of-the-art performance. Further optimization remains an avenue for future work.
>
> Please let us know if you have any additional questions or would like any clarification on any of the above points!
>
> [1] "Robustness verification for transformers." Shi et al. ICLR'20
>
> [2] "Fast and precise certification of transformers." Bonaert et al. PLDI'21
>
> [3] "Beta-CROWN: Efficient bound propagation with per-neuron split constraints for complete and incomplete neural network verification." Wang et al. Neurips'21

---

> > ### Comment · Reviewer_AbHd · 2025-11-26
> > **Response by Reviewer**
> >
> > Thank the authors for posting their response. I read the revised paper and the rebuttal.  However, several of my key concerns remain unaddressed.
> >
> > 1. The central claim of Theorem 4.1 remains tautological. By definition in Eq. 9, $\Delta*$ is the maximum optimal value under perturbation $\le \eta$.  The theorem simply states that if the quantization error (a form of perturbation) is smaller than $\eta$,  the resulting value will be smaller than $\Delta*$. This is a self-evident consequence of the definition, and the proof in Appendix D also follows this directly. I fail to see the novel theoretical insight provided. Furthermore, the authors claim in the rebuttal that "Theorem 4.1 provides the key practical insight that AWP can serve as a differentiable proxy..., enables us to train networks ... under quantization without requiring non-differentiable....". It is overstated. When the perturbation range is large, the method is a kind of well-established fixed-point training. Consequently, Theorem 4.1 does not appear to offer a contribution over existing literature.
> > 2. Although the authors claim they have added a runtime analysis discussion in Section 5.2 and Appendix E. But the experiment about memory cost is not reported in the paper. __The rebuttal's mention that CACTUS consumes 23.7 GB for CNN7 is concerning, especially when considering that a typical ResNet18 (with ~11M parameters) requires far less.__ This suggests an extreme memory overhead that is not properly quantified or discussed in the manuscript.
> > 3. In Section E.1, the authors mention that  "For large networks where memory is a constraint, CACTUS’s loss can be computed for each network in $C(f_\theta)$ separately using gradient accumulation," which will make the training significantly slower (sequential and large memory I/O), which is not practical.
> > 4. The authors' approach of approximating Eq. 6 by using only two models ("the full network and a randomly sampled network") provides a poor and highly unstable estimate. This methodological choice likely undermines the reliability of the certification results presented.
> > 5. Although the author already fixed many notation mistakes in the paper, their current notations still need to be improved, like $Q(f_{\theta}, q_{step})(x) $ will make the reader very confused.
> >
> > Although I think the motivation of addressing certified robustness for model compression is good, the unresolved issues with methodological rigor, practical utility, and presentation clarity lead me to vote for rejection.

---

> > > ### Author Response · Authors · 2025-12-03
> > >
> > > We thank the reviewer for their continued engagement and for taking the time to read our revised paper. We appreciate the opportunity to further clarify the remaining concerns.
> > >
> > > > 1. Theorem 4.1: Theoretical Contribution
> > >
> > > We appreciate the reviewer's concern and would like to provide a more detailed technical justification for why we feel Theorem 4.1 is valuable.
> > >
> > > **The core technical challenge**: CACTUS must jointly optimize for certified robustness across both pruning and quantization. While pruning is differentiable (gradients flow through the subset of active weights), quantization involves non-differentiable rounding operations.
> > >
> > > **Our key insight**: We show that Adversarial Weight Perturbation (AWP) provides a *sound upper bound* on the quantized network's certified loss. This is not simply "optimizing for larger perturbations bounds smaller ones", the technical contribution lies in establishing that the $\ell_\infty$ ball of AWP perturbations *contains* the quantization mapping as a specific point (when $\eta \geq q_{step}/2$). This means:
> > >
> > > 1. **Capturing Certified Loss on Quantized Networks**: When we minimize the AWP-augmented certified loss, we are provably minimizing an upper bound on the certified loss of any quantized network within the covered precision range. This is crucial for capturing the robust certified loss of the quantized network. The theorem gives us a gaurantee on the functional closeness between AWP and quantized networks which is a unique insight that is not explored in previous works to the best of our knowledge.
> > >
> > > 2. **Unified gradient computation**: The CACTUS loss (Eq. 6) requires computing $\nabla_\theta L_{cert} (f_\theta^{Q,q_{step}})$ for quantized networks. Since $Q(\cdot)$ is non-differentiable, we instead compute $\nabla_\theta L_{cert} (f_{\theta+\Delta*})$ where $\Delta^*$ is obtained via gradient ascent on the AWP objective. The theorem guarantees this provides a valid descent direction for the quantized network's loss.
> > >
> > > 3. **Multi-precision coverage**: A single AWP perturbation magnitude $\eta$ covers all quantization step sizes $q_{step} \leq 2\eta$. This enables CACTUS to train for multiple quantization precisions (fp16, int8, etc.) simultaneously without separate forward passes for each precision.
> > >
> > > **Why this is not a straightforward combination**: Prior work on AWP [5] shows that AWP leads to smooth networks that have better empircal robustness. Our contribution is connecting AWP to the *certified training* objective specifically for quantization. The certified loss $\mathcal{L}_{cert}$ involves interval bound propagation (IBP) through the network, and showing that AWP-perturbed bounds dominate quantized bounds requires careful analysis of how weight perturbations interact with IBP's layer-wise bound computation.
> > >
> > > **Practical impact**: Without this theoretical grounding, one might attempt alternatives like:
> > > - Quantization-aware training (QAT): No certified guarantees
> > > - Training on explicitly quantized networks: Non-differentiable, requires separate models per precision
> > > - Ignoring quantization during training: Leads to catastrophic certified accuracy drops (e.g., SABR: 35.13% → 1.12% under int8)
> > >
> > > Our theorem justifies why CACTUS's AWP-based approach maintains certified accuracy under quantization while remaining fully differentiable and covering multiple precisions in a single training run.
> > >
> > > > 2. Memory Overhead Concerns
> > >
> > > We have now added a dedicated section on memory overhead to Section 5.2 of the revised paper. Prior work on certified training reports similar memory requirements: Xu et al. [1] report that IBP for CNN7 requires 7.06 GB, with CROWN-based verification requiring up to 20 GB during training.
> > >
> > > Our measurements are consistent with these prior findings. SABR's 12.8 GB and CACTUS's 23.7 GB represent roughly the expected overhead for adding an additional network to the training process. We will investigate potential memory optimizations and conduct more thorough profiling for the final version of the paper to provide even more detailed breakdowns.

---

> > > > ### Author Response · Authors · 2025-12-03
> > > >
> > > > > 3. Gradient Accumulation Practicality
> > > >
> > > > We acknowledge that gradient accumulation increases training time. However, we respectfully maintain that this does not render CACTUS "not practical" for several important reasons:
> > > >
> > > > First, training with gradient accumulation is still preferable to having no solution for compression-aware certified training. The alternative approaches (post-hoc compression of certified models) suffer catastrophic drops in certified accuracy as demonstrated in our experiments—for example, SABR drops from 35.13% to 1.12% certified accuracy under int8 quantization on CIFAR-10 with $\epsilon$=8/255.
> > > >
> > > > Second, this represents a one-time training cost that can be amortized across multiple deployment scenarios. A single CACTUS model can be dynamically deployed at various compression levels and techniques without retraining, e.g. when compressing for different edge devices [2, 3], making it more efficient than training separate models for each compression configuration.
> > > >
> > > > Third, considering CACTUS's substantial improvements in safety and certified robustness under compression, we believe the additional training time is a worthwhile trade-off for applications where certified guarantees are critical (e.g., safety-critical systems, medical devices, autonomous vehicles).
> > > >
> > > > > 4. Compression Set Approximation: Two Models
> > > >
> > > > We appreciate the reviewer raising this concern. Our ablation studies in Appendix F.3 and F.4 demonstrate that using 2 models per batch (full network + 1 randomly sampled compressed variant) provides a strong balance between performance and computational cost. From Table F.3 in the appendix:
> > > >
> > > > | Strategy | Training Time (min) | 70% Pruned Std. | 70% Pruned Cert. |
> > > > |----------|---------------------|-----------------|------------------|
> > > > | Ours: U(0.25,0.75) (2 models) | 416 | 51.94 | 32.46 |
> > > > | Fixed: [0.25,0.5,0.75] (3 models) | 623 | 51.31 | 32.10 |
> > > > | Dense Sampling (5 models) | 1124 | 51.89 | 32.51 |
> > > >
> > > > And from Table F.4:
> > > >
> > > > | Set Size | 70% Pruned Std. | 70% Pruned Cert. |
> > > > |----------|-----------------|------------------|
> > > > | 2 models (ours) | 51.94 | 32.46 |
> > > > | 3 models | 51.63 | 33.21 |
> > > > | 5 models | 51.78 | 32.67 |
> > > >
> > > > The results show that increasing from 2 to 3-5 models yields marginal improvements (at most 0.75% certified accuracy) while substantially increasing training time and memory overhead. Furthermore, through random sampling over the course of training (100 epochs × ~1,500 batches/epoch = 150,000 batches), the network is exposed to a diverse range of compression configurations, effectively exploring the compression space. This strategy has shown success in other areas of training, i.e. for corruption robustness PRIME [4] shows that augmenting random transformations during training learns a network which is robust to an entire distribution of transformations.
> > > >
> > > > > 5. Notation
> > > >
> > > > We have changed the quantization notation to make it clearer, we have added a paragraph in the background to introduce the new notation and updated the text of the theorem and proof to address this change.
> > > >
> > > > ---
> > > >
> > > > We genuinely appreciate the reviewer's thorough evaluation. We want to emphasize that **we agree with the reviewer that addressing certified robustness for model compression is an important research direction**. We believe that CACTUS takes the first step toward realizing this goal by demonstrating that joint compression-aware training can achieve substantial improvements in certified accuracy under compression compared to sequential approaches. While there are computational trade-offs, we believe the safety benefits, particularly the ability to maintain certified guarantees under compression, represent a meaningful advance for deploying robust models in resource-constrained environments. CACTUS is also effective under a wide range of compressions amortizing the comptuational costs.
> > > >
> > > > We remain committed to incorporating feedback to strengthen the final version and hope our clarifications address the reviewer's concerns.
> > > >
> > > > [1] "Automatic Perturbation Analysis for Scalable Certified Robustness and Beyond"eyond." Xu et al. NeurIPS'20
> > > >
> > > > [2] "EdgeMLBalancer: A Self-Adaptive Approach for Dynamic Model Switching on Resource-Constrained Edge Devices" Matathammal et al. ICSA-C'25
> > > >
> > > > [3] "Dynamic resource allocation for AI/ML applications in edge computing: Framework architecture and optimization methods." Islam et al. JAIGS'24
> > > >
> > > > [4] "PRIME: A few primitives can boost robustness to common corruptions." Modas et al. ECCV'22
> > > >
> > > > [5] "Adversarial Weight Perturbation Helps Robust Generalization." Wu et al. NeurIPS'20

---

### Official Review · Reviewer_7Fvw · 2025-10-31

**Soundness:** 3
**Presentation:** 3
**Contribution:** 3
**Rating:** 8
**Confidence:** 4

**Summary:**

Introduces CACTUS, a joint training framework that optimizes for certified robustness while being compression-aware across pruning and quantization. Core idea: train over a set of compressed variants per batch, with a curriculum on the robust loss and a differentiable proxy for quantization via Adversarial Weight Perturbation (AWP), supported by a bound linking AWP to quantization error. Strong gains appear primarily under compression; uncompressed models remain comparable to robust baselines.

**Strengths:**

- Clear problem formulation unifying certified training with compression; objective over a compression set is well motivated.
- Theory for quantization: a clean reduction from quantization to weight-bounded perturbations via AWP with a formal upper-bound guarantee.
- Consistent empirical gains under compression: across pruning and quantization, CACTUS improves certified accuracy versus robust baselines; integration with multiple certified-training losses shows method generality.
- Ablations beyond the core table: AWP radius sweep, compression-set selection strategies, variance across seeds, extreme sparsity, more bit-widths, additional architectures, and datasets.

**Weaknesses:**

- Scope of headline comparisons: By design, CACTUS is strongest when compressed; for $\delta$=0 or unquantized, SABR typically wins. This is expected but should be emphasized alongside deployment guidance.

- Condition discrepancy: Main text, Theorem 4.1 states $q_{step}\leq \eta$ while Appendix Theorem D.1 states $q_{step}\leq 2\eta$. The bound is fine, yet the precise requirement should be consistently stated.

- Compute cost: Training time overhead is non-trivial; although addressed in Appendix E, a compact cost-vs-benefit summary in the main paper would help.

**Questions:**

- Uncompressed use case: For δ=0 or no quantization, is there a recommended $\lambda$ schedule or training variant that narrows the remaining gap to SABR, or is the intended operating point strictly under compression
- The appendix covers 0.9–0.99 sparsity and multiple bit-widths. Could the main paper include a compact figure summarizing these extremes to highlight how CACTUS compares against baselines, even under extreme sparsity?
- Could there be more clarity provided on the aforementioned "Condition discrepancy" (Weaknesses)?
- Maybe consider citing [1], as it directly studies robustness effects under compression and would contextualize CACTUS relative to prior compression-robustness efforts for robustness other than adversarial and certified robustness.
- Would it be possible to add something like a "Table of Contents" in the Appendix, before the content starts to clearly list all the additional studies, experiments, and other contents in the Appendix, since including a portion of this information in the "Reproducibility Statement" does not clearly state all the experiments in the Appendix.

References:

[1] Hoffmann, J., et al. "Towards improving robustness of compressed CNNs." ICML Workshop on Uncertainty and Robustness in Deep Learning (UDL). 2021.

---

> ### Author Response · Authors · 2025-11-21
>
> We thank Reviewer 7Fvw for the positive evaluation and for recognizing CACTUS's "clear problem formulation," "clean theory," "consistent empirical gains," and comprehensive ablations. We will address all concerns raised to further strengthen the paper.
>
> > Scope of Headline Comparisons
>
> We have emphasized this point in Section 1 (Introduction).
>
> > Condition Discrepancy in Theorem 4.1
>
> Thank you for catching this inconsistency. You are correct, and we apologize for the confusion. The correct statement is $q_{step} \leq 2\eta$ as stated in Appendix Theorem D.1, and we have corrected the typo in Theorem 4.1 in Section 4 (Technical Contribution) to match this.
>
> > Compute Cost Prominence
>
> We have added a compact summary in Section 5 (Evaluation) with a reference to the detailed discussion in Appendix D (Experimental Details).
>
> > Uncompressed Performance Gap to SABR
>
> This is by design but should be emphasized more clearly, as you note. CACTUS is designed for certified accuracy under compression. We believe that given CACTUS's increased optimization complexity, it achieves performance that is on par with SABR while uncompressed, but do note that CACTUS's intended use case is for dynamic compression scenarios where the network is mostly utilized in a compressed manner. We have made this point clearer in Section 1 (Introduction) and Section 5 (Evaluation).
>
> > Extreme Sparsity Figure in Main Paper
>
> Excellent suggestion. We have results for 0.9, 0.95, 0.99 sparsity in Appendix F.5 but agree a compact visualization would strengthen the main paper. We have added to Section 5 (Evaluation) a figure showing CACTUS versus SABR certified accuracy across sparsity levels [0, 0.5, 0.7, 0.9, 0.95, 0.99] for CIFAR-10. This makes the paper's claims about extreme compression more immediately visible without requiring readers to search the appendix.
>
> > Additional Citation and Appendix Organization
>
> Thank you for the Hoffmann et al. 2021 reference. We have added this citation to Section 3 (Related Work). We also agree that the appendix would benefit from better organization, and have added a detailed table of contents at the beginning of the appendix.
>
>
> Please let us know if you have any additional questions or would like any clarification on any of the above points!

---

> > ### Comment · Reviewer_7Fvw · 2025-11-26
> >
> > My concerns have been addressed.
> >
> > For now, I will maintain my high score.
> >
> > Though I am curious about the concerns of Reviewer AbHd.
> >
> > I will make a final decision based on how that conversation flows.
> >
> > Best Regards

---

### Official Review · Reviewer_Fxhk · 2025-11-01

**Soundness:** 3
**Presentation:** 2
**Contribution:** 2
**Rating:** 4
**Confidence:** 3

**Summary:**

The paper introduces CACTUS (Compression-Aware Certified Training Using Network Sets), a framework designed to jointly train neural networks to be both certifiably robust and compressible through pruning and quantization. The approach integrates a joint objective that combines certified robustness and compression losses across a set of compressed model variants. To approximate the effects of quantization in a differentiable manner, CACTUS employs Adversarial Weight Perturbation (AWP), supported by Theorem 4.1, which formally links AWP bounds to quantization error. Empirical evaluations on MNIST, CIFAR-10, and TinyImageNet demonstrate that CACTUS achieves higher certified accuracy compared to post-hoc compression baselines such as SABR, HYDRA, NRSLoss, and QA-IBP. The authors claim that CACTUS represents the first unified framework that integrates compression and certified robustness within a single training process.

**Strengths:**

Deploying robust models on resource-limited devices is an important and timely research direction.

The joint training objective is clearly defined and implemented. The use of compression sets and curriculum-based loss weighting is technically reasonable.

Experiments are carefully executed and include ablations (AWP radius, compression-set size). CACTUS consistently outperforms sequential baselines in certified accuracy under compression.

The paper is well written, equations are clean, and implementation details are fully specified.

**Weaknesses:**

The work overlooks Gui et al. (2019), "Model Compression with Adversarial Robustness: A Unified Optimization Framework", which already introduced a unified optimization framework combining model compression (pruning and quantization) with adversarial training. While ATMC focused on empirical rather than certified robustness, the underlying idea (joint optimization of robustness and compression) is the same. A clearer connection to this prior line of work would strengthen the paper’s positioning and clarify its contribution.

Results are restricted to small and mid-scale datasets (MNIST, CIFAR-10 and TinyImageNet).

Although the paper briefly reports results for joint pruning and quantization (Table 5), these results are weaker and not deeply analyzed. A deeper discussion on why multi-objective CACTUS performs suboptimally would be valuable.

**Questions:**

How does CACTUS differ formally from ATMC beyond using certified losses instead of adversarial training losses?

Could the CACTUS framework be applied to text or speech models, where quantization effects differ significantly?

Would combining ATMC’s constrained optimization with certified bounds yield similar or better results?

---

> ### Author Response · Authors · 2025-11-21
>
> We thank Reviewer Fxhk for acknowledging that "deploying robust models on resource-limited devices is an important and timely research direction" and that our "experiments are carefully executed" with CACTUS "consistently outperforming sequential baselines." We will leverage your feedback to strengthen our positioning and expand our analysis.
>
> > Related Work: ATMC (Gui et al., 2019)
>
> We sincerely thank the reviewer for bringing this important reference to our attention. While ATMC [1] and CACTUS share the high-level goal of joint optimization for compression and robustness, they differ fundamentally in their objectives, guarantees, and technical approaches. We note the following key differences:
>
> - **Compression Set Size**: ATMC defines their optimization function (Eq. 4) over two hyperparameters: $k$ for sparsity and $b$ for quantization precision, i.e., ATMC trains a sparse, quantized network. CACTUS's objective function (Eq. 6) aims to learn a network that is performant over a wide range of compression techniques and levels, i.e., CACTUS trains a normal network which can later be compressed while maintaining accuracy and provable robustness depending on runtime needs. This allows a single trained CACTUS network to be useful at multiple compression levels and under different compression techniques.
> - **Compression Type**: ATMC designs their own quantization and pruning framework to optimize over. CACTUS is posed as a general framework which can adapt to a variety of commonly used techniques (global/local, $l_2$/$l_1$ pruning, fp16/int8 quantization, etc.). This allows CACTUS trained networks to be more easily adapted into diverse environments.
> - **Certified Training Objective**: ATMC focuses on empirical adversarial training, while CACTUS focuses on certified robustness, which is a harder problem to optimize.
>
> These differences are reflected in CACTUS's design choices and theoretical studies: (1) we introduce Theorem 4.1, which provides theoretical guarantees on using Adversarial Weight Perturbation as a differentiable proxy for quantization, (2) in our ablation studies, we also explore compression set selection strategies and analyze how set size and selection affect performance (Appendix D.3).
>
> We have revised Section 3 (Related Work) to add ATMC with a thorough comparison highlighting these differences, and updated Section 1 (Introduction) to reframe our contribution more precisely as "first framework for joint compression-aware certified training" rather than claiming to be the first joint optimization approach more broadly, including a discussion of how certified robustness requires fundamentally different techniques than empirical robustness.
>
> > Limited to Small/Mid-Scale Datasets
>
> We acknowledge this limitation and have added clarification to Appendix G (Limitations). While we agree that evaluation on larger-scale datasets would strengthen our work, we note that CACTUS evaluates on the same scale of networks and datasets used by existing works on certified training [2,3]. In Appendix F.1, we discuss how CACTUS's contributions are in parallel with advances in certified training, i.e., as new certified training methods develop and scale to larger networks this allows CACTUS to scale to larger networks as well. In particular, recent works have shown that IBP-based verification can scale to transformer networks [4,5].
>
> > Joint Pruning + Quantization Results
>
> The joint optimization results (Table 5) show that while the jointly trained model does not match specialized models, it achieves better quantized performance than the pruned model and better pruned performance than the quantized model. Under a scenario where a model might be compressed using different techniques, the CACTUS model trained with both strikes a balance, achieving good performance on both. We believe that optimizing for both quantization and pruning is inherently more difficult. Networks optimized for pruning tend to concentrate representational capacity into a small subset of important weights [7], while networks optimized for quantization prefer weights clustering near quantization levels [8] or oscillating around quantization thresholds [9]. Simultaneously considering both objectives increases the optimization complexity. We have extended this discussion in Section 5 (Evaluation).
>
> > Could CACTUS be applied to text or speech models?
>
> Yes, CACTUS's framework is general in that it defines a way to optimize certified accuracy under a range of compression techniques. As mentioned above, existing work has shown IBP can be scaled to transformer networks [4,5], and as techniques for certified training for LLMs/Speech Networks appear, CACTUS's framework can adapt to those methods. CACTUS's theoretical analysis of AWP as a differentiable proxy for quantization is not model architecture/domain specific and would therefore extend to these new domains as well.

---

> ### Author Response · Authors · 2025-11-21
>
> > Would combining ATMC's constrained optimization with certified bounds yield similar results?
>
> ATMC's optimization approach consists of the following steps: 1. compute adversarial example, 2. optimize weights for adversarial example while constrained to sparsity level, 3. shift weights to closest quantized weight equivalent using clustering. Mimicking these steps, we implemented constrained optimization with certified bounds. For each batch, we first update the network using the certified training loss (we start with standard loss and add IBP loss with increasing $\epsilon$ as done in CACTUS and other certified training works), then project the weights under the sparsity condition ($|\theta|_0 \leq k$). Next, as described in the ATMC paper, we compute the cluster centers (including a constant cluster 0) with the new weights. Finally, we move each weight to its cluster center.
>
> For our experiment, we use CIFAR-10 with 70% sparsity, a bit width of 16, and $\epsilon = 8/255$. ATMC with certified bounds achieves a standard accuracy of 34.21 and a certified accuracy of 18.64. We instantiate CACTUS using a compression set with global unstructured $l_1$ pruning at 70% and fp16 quantization. When pruned and quantized, CACTUS achieves a standard accuracy of 40.18 and a certified accuracy of 28.35. We note that CACTUS achieves much better performance than ATMC with certified bounds. We believe that CACTUS achieves this performance gain as it performs simultaneous optimization of pruning and quantization, whereas ATMC sequentially first prunes and then quantizes its network, leading to suboptimal performance. Furthermore, CACTUS's use of AWP to approximate quantization also allows us to integrate 'quantization' effects into the loss function. In the final version of the paper, we will explore constrained optimization in more detail and include this as an ablation study.
>
> Please let us know if you have any additional questions or would like any clarification on any of the above points!
>
> [1] "Model Compression with Adversarial Robustness: A Unified Optimization Framework." Gui et al. Neurips'19
>
> [2] "Certified Training: Small Boxes are All You Need." Müller et al. ICLR'23
>
> [3] "Connecting Certified and Adversarial Training." Mao et al. ICML'23
>
> [4] "Robustness verification for transformers." Shi et al. ICLR'20
>
> [5] "Fast and precise certification of transformers." Bonaert et al. PLDI'21
>
> [6] "Exploiting the Partly Scratch-off Lottery Ticket for Quantization-Aware Training." Zhong et al.
>
> [7] “Learning Both Weights and Connections for Efficient Neural Networks.” Han et al. NIPS'15.
>
> [8] “Cluster-Promoting Quantization.” Lee et al. ICCV'21.
>
> [9] “Overcoming Oscillations in Quantization-Aware Training.” Nagel et al. ICML'22.

---

### Author Response · Authors · 2025-11-21

We would like to thank the reviewers for their detailed and thoughtful feedback! We have made a number of edits to the new version of the paper, which we believe have addressed the reviewer's feedback and increased the paper quality. Changes to the PDF are in blue so they are easier to see.

---

### Author Response · Authors · 2025-12-03

We sincerely thank all reviewers and the Area Chair for the thoughtful evaluation and constructive feedback on our work. We have carefully addressed each concern raised and believe the revised paper is significantly strengthened as a result.

## Strengths Highlighted by Reviewers

The reviewers recognized several key contributions of CACTUS:

- **Important and Timely Problem**: Reviewers acknowledged that "deploying robust models on resource-limited devices is an important and timely research direction" (Fxhk) and that CACTUS is "the first work to address certified robustness for model compression" (AbHd).

- **Clear Technical Formulation**: The paper presents a "clear problem formulation unifying certified training with compression" with a "well motivated" objective over compression sets (7Fvw). The "joint training objective is clearly defined and implemented" with "technically reasonable" use of compression sets and curriculum-based loss weighting (Fxhk).

- **Strong Theoretical Foundation**: Reviewers noted the "clean theory" providing "a clean reduction from quantization to weight-bounded perturbations via AWP with a formal upper-bound guarantee" (7Fvw).

- **Consistent Empirical Gains**: CACTUS "consistently outperforms sequential baselines in certified accuracy under compression" (Fxhk), with "consistent empirical gains under compression across pruning and quantization" and "integration with multiple certified-training losses shows method generality" (7Fvw).

- **Comprehensive Evaluation**: "Experiments are carefully executed" with thorough ablations covering "AWP radius sweep, compression-set selection strategies, variance across seeds, extreme sparsity, more bit-widths, additional architectures, and datasets" (Fxhk, 7Fvw).

- **Strong Presentation**: "The paper is well written, equations are clean, and implementation details are fully specified" (Fxhk), and "the background section is well-written, providing a clear foundation" (AbHd).

## Main Revisions in Response to Reviewer Feedback

We have made substantial revisions to address all reviewer concerns:

1. **Related Work and Positioning**: Added thorough comparison with ATMC (Gui et al., 2019) in Section 3 and reframed our contribution as "first framework for joint compression-aware certified training" to more precisely position our work (Fxhk).

2. **Notation and Presentation**: Comprehensively fixed all notation issues including consistent bold font for vectors, clear $f_\theta$ definition, corrected $\Delta^*$ usage in Theorem 4.1, and fixed LaTeX syntax errors (AbHd).

3. **Theorem Condition Consistency**: Corrected the discrepancy between main text and appendix, with the correct condition being $q_{step} \leq 2\eta$ now stated consistently throughout (7Fvw, AbHd).

4. **Computational Overhead Analysis**: Added runtime and memory overhead discussion in Section 5.2, including specific measurements (33-40% training time increase, 85% memory increase) with reference to detailed analysis in Appendix E (AbHd, 7Fvw).

5. **New Ablation Studies**: Added comprehensive ablations on $\lambda$ schedule and upper bound values in Appendix C.2, with clear justification for our design choices (AbHd).

6. **Extreme Sparsity Visualization**: Added a figure in Section 5 showing CACTUS vs. SABR certified accuracy across sparsity levels [0, 0.5, 0.7, 0.9, 0.95, 0.99] to make extreme compression claims immediately visible (7Fvw).

7. **Appendix Organization**: Added a detailed table of contents at the beginning of the appendix for improved navigation (7Fvw).

8. **Additional Citations**: Added Hoffmann et al. (2021) reference to contextualize CACTUS relative to prior compression-robustness work (7Fvw).

9. **Clarified Experimental Details**: Explicitly documented certified accuracy computation using $\alpha,\beta$-CROWN verifier in the experimental setup (AbHd).

10. **Extended Discussion**: Expanded analysis of joint pruning + quantization results and clarified CACTUS's intended operating point under compression (Fxhk, 7Fvw).

## Conclusion

We believe these revisions have substantially strengthened the paper. We have addressed all technical concerns raised by the reviewers, improved the clarity and consistency of our presentation, and provided additional experimental evidence supporting our claims. CACTUS represents an important step toward deploying certifiably robust models in resource-constrained environments, and we are confident that the revised manuscript now clearly communicates both the significance and the technical contributions of our work. We thank the reviewers and the Area Chair for their time and valuable feedback, and we believe the revised paper addresses all raised concerns.

---

### Meta-Review · Area_Chair_7DFH · 2026-01-06

**Summary:**

This paper introduces a framework called CACTUS to jointly train neural networks to achieve certified robustness while being compression-aware across pruning and quantization. Empirical evaluations on MNIST, CIFAR-10, and TinyImageNet demonstrate the resulting models have higher certified accuracy compared to post-hoc compression baselines. While the topic is important and the proposed framework is interesting, there are some unresolved concerns from the reviewers, including results available only on relatively small scale datasets and training overhead on memory/speed, and the presentation. Hence a rejection is recommended. The authors are encouraged to address the reviewers concerns to strengthen the manuscript.

**Reviewer Concerns:**

* Reviewer Fxjk's concern on overlapping idea with existing work has been addressed. However, the reviewer's concern on results only on small and mid-scale dataset is not addressed.

* Reviewer 7Fvw's concerns are addressed but stated will make final decision based on author discussion between Reviewer AbHd.

* Reviewer AbHd stated there are still unresolved issues on the claim of Thm 4.1, extreme memory overhead, concern on training efficiency and Eq 6 approximation.

**Reviewer Scores:**

* Reviewer Fxjk will likely remain score of 4  or increase to 5
* Reviewer 7Fvw will likely remain score of 8 or downgrade to 6
* Reviewer AbHd stated there are still unresolved issues with methodological rigor, practical utility, and presentation clarity and hence vote for rejection. Hence Reviewer AbHd will likely remain the score of 2.

---

### Decision · Program_Chairs · 2026-01-26

Reject